

# Impact of Lagrangian Transport on Lower-Stratospheric Transport Time Scales in a Climate Model

Edward J. Charlesworth[1], Ann-Kristin Dugstad[1], Frauke Fritsch[2], Patrick Jöckel[2], and Felix Plöger[1,3]

[1]Forschungszentrum Jülich, IEK-7 Stratosphäre, Germany
[2]Deutsches Zentrum für Luft- und Raumfahrt, Institut für Physik der Atmosphäre, Germany
[3]Institut für Atmosphären- und Umweltforschung, Universität Wuppertal, Germany

**Correspondence:** Edward J. Charlesworth (Edward.Charlesworth.Science@GMail.com)

**Abstract.** We investigate the impact of model trace gas transport schemes on the representation of transport processes in the upper troposphere and lower stratosphere. Towards this end, the Chemical Lagrangian Model of the Stratosphere (CLaMS) was coupled to the ECHAM/MESSy Atmospheric Chemistry (EMAC) model and results from the two transport schemes were compared. Advection in CLaMS was driven by the EMAC simulation winds and thereby the only differences in transport between

the two sets of results were caused by differences in the transport schemes. To analyze the time scales of large-scale transport, multiple tropical-surface-emitted tracer pulses were performed to calculate age of air spectra, while smaller-scale transport was analyzed via idealized, radioactively-decaying tracers emitted in smaller regions (nine grid cells) within the stratosphere. The results show that stratospheric transport barriers are significantly stronger for Lagrangian EMAC-CLaMS transport due to reduced numerical diffusion. In particular, stronger tracer gradients emerge around the polar vortex, at the subtropical jets, and

at the edge of the tropical pipe. Inside the polar vortex, the more diffusive EMAC flux-form semi-Lagrangian transport scheme results in a substantially higher amount of air with ages from 0 to 2 years (up to a factor 5 higher). In the lowermost stratosphere, air is much younger in EMAC, owing to stronger diffusive cross-tropopause transport. Conversely, EMAC-CLaMS shows a summertime lowermost stratosphere age inversion - a layer of older air residing below younger air (an "eave"). This pattern is caused by strong poleward transport above the subtropical jet, and is entirely blurred by diffusive cross-tropopause

transport in EMAC. Potential consequences from the choice of the transport scheme on CCM and geoengineering simulations are discussed.

## 1 Introduction

The upper troposphere and lower stratosphere (UTLS) is an important region for global climate as the chemical composition of radiatively active trace gas species there has crucial impacts on radiation and surface temperatures (e.g., Solomon et al., 2010).

Understanding the UTLS, however, is complicated by the fact that the entry of air masses into the stratosphere is controlled by the chemical and dynamical processes in the UTLS (e.g., Holton et al., 1995; Fueglistaler et al., 2009). To overcome this challenge, climate models must have a realistic representation of UTLS transport processes in order to provide reliable predictions and assist in robust theoretical development. In particular, for simulating the effects of geoengineering by sulfur injections into the stratosphere, uncertainties in the model transport representation could cause substantial uncertainties in the simulations





(Tilmes et al., 2018; Kravitz and Douglas, 2020). Even small differences in composition caused by model differences in small-scale transport processes (e.g., turbulence, diffusion) may cause significant model spread in surface temperatures (e.g., Riese et al., 2012). This radiative effect of composition changes in the upper troposphere and lower stratosphere is particularly large for water vapour, but also substantial for other species like $O_3$, $N_2O$, and $CH_4$.

The representation of transport processes in the lower stratosphere in global models is prone to numerical diffusion, as tracer distributions in this region are characterized by sharp gradients and frequent small-scale filamentary structures (McKenna et al., 2002). Critical processes for models are transport around the wintertime stratospheric polar vortex, stratosphere-troposphere exchange across the tropopause, and horizontal exchange between the tropical lower stratosphere (the tropical pipe, Plumb, 1996) and middle latitudes (for reviews of stratospheric transport processes see e.g., Plumb, 2002; Shepherd, 2007). The steep

gradients in observed trace gas distributions in these regions are signs of transport barriers and regions of suppressed exchange, for example, around the polar vortex, at the edge of the tropical pipe, and along the extratropical tropopause.

In general, different atmospheric models (as used in current coupled chemistry climate models) employ different numerical schemes for solving trace gas transport, all of which introduce some unwanted, unphysical numerical diffusion. Numerical diffusion smoothes gradients and small-scale filaments in tracer distributions, and thereby differences in numerical diffusion

cause differences in trace gas transport in different models, affecting the simulated distributions of trace gas species (e.g., Eluszkiewicz et al., 2000; Gregory and West, 2020)

Most currently-used transport schemes are based on a regular grid (e.g., Morgenstern et al., 2010) and will be referred to as Eulerian schemes in the text to follow. Lagrangian schemes, on the other hand, follow the motion of air parcels through the atmospheric flow, and hence have reduced diffusion characteristics due to the absence of interpolations of tracer distributions

to a regular grid (e.g., McKenna et al., 2002). Semi-Lagrangian schemes are still based on a regular grid, but incorporate some advantages of Lagrangian transport by calculating the air motion over one model time step through a Lagrangian advection scheme, but this is then followed by remapping onto the grid. One such scheme which is both sophisticated and frequently used in global models is the flux-form semi-Lagrangian (FFSL) scheme (e.g., Lin and Rood, 1996; Lin, 2004).

Fully Lagrangian transport schemes, by definition, are free of numerical diffusion, as parcels are left entirely isolated from

each other when no inter-parcel mixing scheme is applied. Parcel mixing due to small-scale processes (e.g., turbulence) can then be introduced based on physical parameterizations and the strength of mixing can then be controlled. Due to the complications of handling irregular (air parcel) grids, Lagrangian schemes are not commonly used in global climate models To our knowledge, the only two Lagrangian transport schemes which are currently implemented in a global climate model are ATTILA (Stenke et al., 2008, 2009; Brinkop and Jöckel, 2019). and CLaMS (Hoppe et al., 2014, 2016). Both these schemes have been integrated

into the ECHAM/MESSy Atmospheric Chemistry (EMAC) climate model (e.g., Jöckel et al., 2005, 2016) and at the present time neither has been incorporated into another climate model.

Stenke et al. (2008) showed that using the ATTILA scheme in EMAC reduced the excessive transport of water vapour into the lowermost stratosphere and into polar regions and the associated cold bias in temperatures could be partly corrected. The representation of stratospheric ozone was also found to have been improved (Stenke et al., 2009). Hoppe et al. (2014) further



showed that CLaMS transport within EMAC results in a more realistic representation of transport barriers around the southern polar vortex, due to reduced numerical diffusion compared to the EMAC FFSL scheme.

Here, we build on the study of Hoppe et al. (2014) and further analyse the implementation of the Lagrangian transport scheme CLaMS within the EMAC climate model. We compare results from two tracer sets within one EMAC simulation: one

set where transport is calculated using the EMAC FFSL scheme and one set using the CLaMS Lagrangian tracer transport scheme. To enable a more detailed analysis of composition and transport time scales, going beyond the average stratospheric transit time (the mean age, Waugh and Hall, 2002) as considered by Hoppe et al. (2014), we calculated the full (time-dependent) stratospheric age of air spectrum (the distribution of stratospheric transit times) of model transport schemes.

This work investigates the differences in transport in the lower stratosphere between these two transport schemes using the

age spectrum, mean age, and idealized tracers as diagnostics. Questions raised in this work are: (i) Which regions are most critical regarding changes in the tracer transport scheme?; (ii) On which time scales is the model transport differing in different regions?; (iii) What are potential consequences for simulated chemical composition and geoengineering simulations?

In Section 2.1 the used models and diagnostic methods (age spectrum, forward tracers) are introduced. Section 3 presents the results from a global perspective, while Section 4 focuses on particular processes and regions. In Section 5 the transport scheme

differences are discussed against the background of current research on stratospheric geoengineering. The main conclusions are summarized in Section 6.

## 2   Methods

### 2.1   Models

The model used in this work is EMAC, the MESSy (Modular Earth Submodel System) version of the ECHAM5 climate

model (see Jöckel et al. (2010) for details on EMAC and Roeckner et al. (2006) for details on ECHAM5). EMAC is a modern chemistry-climate model which is commonly used for studies of the stratosphere and upper troposphere (Sinnhuber and Meul, 2015; Oberländer-Hayn et al., 2016; Fritsch et al., 2019), as well as studies of the troposphere. In this work, EMAC is operated at the T42L90MA spectral resolution, corresponding to a horizontal quadratic Gaussian grid of approximately 2.8° x 2.8° resolution with 90 vertical layers. One simulation is performed with this model, by which two sets of time-resolved

tracer distributions were calculated. One tracer set was calculated with the standard EMAC FFSL transport scheme, and will be referenced as the Eulerian representation or EMAC-FFSL. The other tracer set was calculated with the CLaMS EMAC submodel, and will be referenced as the Lagrangian representation or EMAC-CLaMS.

The EMAC-FFSL transport scheme is the flux-form semi-Lagrangian (FFSL) scheme (Lin and Rood, 1996), which is used in many modern climate models. The EMAC-FFSL vertical coordinate is a hybrid sigma-pressure coordinate, which is another

common choice in the development of modern climate models. The time resolution of the EMAC simulation performed in this work is 12 minutes. The simulation consists of ten years of spin-up, with a following ten years of result production. The EMAC version used in this work is 2.53.1. Although EMAC can be used for chemistry-climate model simulations, the configuration in this work did not simulate interactive chemical fields. The water vapor field, however, was interactive, and





included stratospheric moistening via methane oxidation (see e.g., Revell et al., 2016). Sea-surface temperatures and and sea ice were prescribed from the HadISST climatology (Rayner et al., 2003). Meanwhile, $CO_2$, $CH_4$, $N_2O$, CFC-11, and CFC-12 mixing ratios were fixed at 367 ppmv, 175 ppmv, 316 ppbv, 262 pptv, and 520 pptv, respectively, for calculation of radiation.

    CLaMS (the Chemical Lagrangian Model of the Stratosphere) is a Lagrangian chemical transport model based on three-
dimensional trajectories and an additional mixing parameterization. The EMAC-CLaMS results in this work were produced with a resolution of approximately 3 million air parcels. Unique among Lagrangian models, CLaMS uses a mixing param- eterization which is robustly based on physical principles. This parameterization is based on the critical Lyapunov exponent method, details of which can be found in Konopka et al. (2004). The vertical coordinate of CLaMS is a hybrid $\sigma - \theta$ coordinate (referred to as $\zeta$) (Hoppe et al., 2014). Above the prescribed reference pressure of 300 hPa, $\zeta$ is identical to $\theta$ and therefore the
vertical advection velocity throughout the stratosphere is identically the diabatic heating rate. CLaMS advection is normally driven by horizontal winds and diabatic heating rates from reanalyses (e.g., Konopka et al., 2007; Ploeger et al., 2019), however in EMAC-CLaMS advection of CLaMS parcels is driven by the horizontal winds and heating rates of EMAC. This advection is driven online, during execution of the simulation, so that the underlying velocity fields for advection in EMAC-CLaMS and EMAC-FFSL are exactly the same. However, there are two differences in how these fields are used by the transport schemes:
(1) EMAC-CLaMS interpolates the horizontal winds onto parcel locations, whereas EMAC-FFSL uses the winds directly on the EMAC grid points; (2) As mentioned above, the vertical velocity of EMAC-CLaMS is the diabatic heating rate (calculated by EMAC), whereas EMAC-FFSL uses a kinematic vertical velocity (calculated by closure of the mass balance equation). The horizontal and vertical velocities in the two transport schemes are therefore consistent, but not actually identical. More details of EMAC-CLaMS are described by Hoppe et al. (2014) and Hoppe et al. (2016).

## 2.2  Age Spectra

The goal of this work is examination of differences in tracer transport between two advection schemes, for which analysis of passive tracers is ideal. This approach, as opposed to examination of chemically-active species, eliminates differences that could arise through the differing chemical schemes of EMAC and and the CLaMS submodel of EMAC. The diagnostic tool used in this work is the age spectrum, $G(r,t,\tau)$, which describes the probability distribution of stratospheric transit times $\tau$
(age) within an air parcel sampled at location $r$ and time $t$ (e.g., Waugh and Hall, 2002). The first moment of the age spectrum is the mean age $\Gamma$, which represents the average transit time from a tracer source region to a given point in the atmosphere

$$\Gamma(\boldsymbol{r},t) = \int\limits_0^\infty \tau\, G(\boldsymbol{r},t,\tau)\,\mathrm{d}\tau\,. \tag{1}$$

In models, age spectra can be calculated by a series of tracers which are pulsed at some reference location (in this case the
tropical surface). For such a tracer with a pulse in the source region at time $t_i$ the mixing ratio $\chi^i(\boldsymbol{r},t)$ at point $\boldsymbol{r}$ and time $t$ can be normalized to the probability density for air of the transit time $\tau = t - t_i$, which is the value of the age spectrum.

$$G(\boldsymbol{r},t,t-t_i) = \chi^i(\boldsymbol{r},t)\,. \tag{2}$$





Therefore, a suite of pulse tracers provides the full transit time dependency of the age spectrum function $G$.

This boundary impulse response method has been used in a few other modelling studies to calculate fully time-dependent stratospheric age spectra (for further details see e.g. Li et al., 2012; Ploeger and Birner, 2016; Hauck et al., 2019). In this work, the tracers are emitted over the course of thirty days, after which emissions are ceased, and one tracer is pulsed every three

months, specifically in January, April, July, and October of each year, analogous to the set-up by Hauck et al. (2019). Tracer emission is performed by prescribing the surface boundary mixing ratio in EMAC. Each tracer is therefore assigned an age based on when the tracer was emitted, and the combined set of tracers is used to create the age distribution. Forty tracers are utilized in total, such that the calculated age spectra span the course of ten years. After 10 years, mixing ratios of the oldest tracer are set to zero throughout the model domain and the tracer is re-pulsed, so that the age spectra always spans from 0 to

10 years. Furthermore, the spectra are normalized so that the integral of the spectra over transit time always equals one.

Due to the truncation of the age spectrum at 10 years of age, although a "true" age spectrum would show a significant fraction of air older than 10 years, the mean age is biased young. This fact is important to bear in mind in comparing the mean age described here to calculations in other studies. It has been shown that the age spectrum tail can be extrapolated to infinity by fitting an exponential decay (e.g., Diallo et al., 2012) and the mean age can be corrected accordingly. However, to facilitate

comparison between EMAC-FFSL and EMAC-CLaMS transport, we refrain from applying this tail correction and focus on the resolved part of the age spectrum with transit times younger than 10 years. The uncalculated differences in the spectrum tail at ages older than 10 years are likely small compared to the differences in the resolved section of the spectra.

## 2.3 Forward Tracers

One disadvantage of the analysis of age spectra is abstraction of results away from the transport of realistic chemically-active

species, such as water and ozone. In the results that follow, considerable differences are found in age spectra between the two considered transport schemes. These results indicate distinct differences in tracer transport, but do not directly predict contrasts in the transport of specific, chemically-active tracers. We therefore investigate additional idealized trace gas species to reflect the results in a less abstract form. In particular, we consider the case of tracers with the simplest chemistry possible - that of radioactive decay. By convoluting an air parcel's age spectrum with an exponentially-decaying weighting, the fraction of a

hypothetical radioactive tracer with a decay lifetime $T$ that would remain after transport from the tropical surface (the origin of the pulse tracers) can be calculated

$$\chi^T(r,t) = \int_0^\infty \chi_0^T \, G(r,t,\tau) \, e^{-\frac{\tau}{T}} \, d\tau \ .$$
(3)

Here, $\chi_0^T$ is the tracer mixing ratio at the tropical surface.

Throughout this paper, this quantity will be referred to as a "forward tracer", as it is computed forward from the knowledge

of the age distributions throughout the model domain.





## 2.4 The EMAC-CLaMS Lower and Upper Boundary

A critical decision in this study lies in the way in which age tracers are pulsed. Differences in the age spectra between the two transport schemes would ideally stem only from differences in transport within the region of interest (the stratosphere and upper troposphere). As mentioned in the introduction, the two transport schemes differ greatly in the representation of convective transport, as EMAC-CLaMS does not account for parameterized convection, while in the grid-point representation the tracers are subject to a convective transport parameterization. To elimate the effects of this difference below the upper troposphere, the age tracer concentrations of the EMAC-CLaMS representation were fixed to those of EMAC-FFSL below level 73 of the EMAC model. This level corresponds to 270 hPa (330 K) in the tropics and extratropics, and about 250 hPa (300 K) in the winter polar region (poleward of 75 degrees). The precise mechanism is as follows: for each EMAC-CLaMS parcel at each timestep, the EMAC grid cell containing the parcel was identified and if the parcel was located at or below EMAC level 73, the EMAC-CLaMS parcel age tracer values were replaced by EMAC-FFSL age tracer values of that EMAC cell. In this way, EMAC-FFSL results do not qualitatively differ from those of EMAC-CLaMS below EMAC level 73 (the upper troposphere). There are, however, small quantitative differences between the two sets of transport scheme results due to interpolation and numerics because the two representations have different grids and resolutions in this region. This creates very minor differences which are most noticable near the surface.

The model top in EMAC is at 0.01 hPa (approximately 80 km) (Jöckel et al., 2016). As the CLaMS transport scheme has not been extended into the mesosphere so far, the uppermost level in EMAC-CLaMS results is around the stratopause (around 2500 K, see Hoppe et al., 2014). Therefore, in regions of downwelling air from the mesosphere, EMAC-CLaMS age of air will be young-biased compared to the EMAC-FFSL age. However, as this paper focuses on the lower stratosphere, the effect of these differences is expected to be weak. Furthermore, as the EMAC-CLaMS age is found to be generally older than the EMAC-FFSL age in the lower stratosphere (see Figure 1), these age differences can be regarded as conservative estimates of inter-representation differences.

## 3 Differences in the Zonal Mean State: Global Perspective

### 3.1 Mean age of air

Examination of mean age of air (in Figure 1) shows many qualitative similarities between the Lagrangian and Eulerian frameworks but also shows substantial quantitative differences and one notable qualitative discrepancy. In both representations, mean age gradually increases with distance from the tropical tropopause layer (TTL), the region from 355–425 K through which most tropospheric air entering the stratosphere passes (e.g., Holton et al., 1995; Fueglistaler et al., 2009; Butchart, 2014). At all potential temperature levels, mean age is lowest in the tropical stratosphere (tropical pipe, Plumb, 1996) and gradually increases towards high latitudes. Mean age is generally lower in the winter than the summer, consistent with stronger wintertime downwelling in the polar region (bringing older air from higher to lower levels) and the isolation of the polar vortex (which limits the





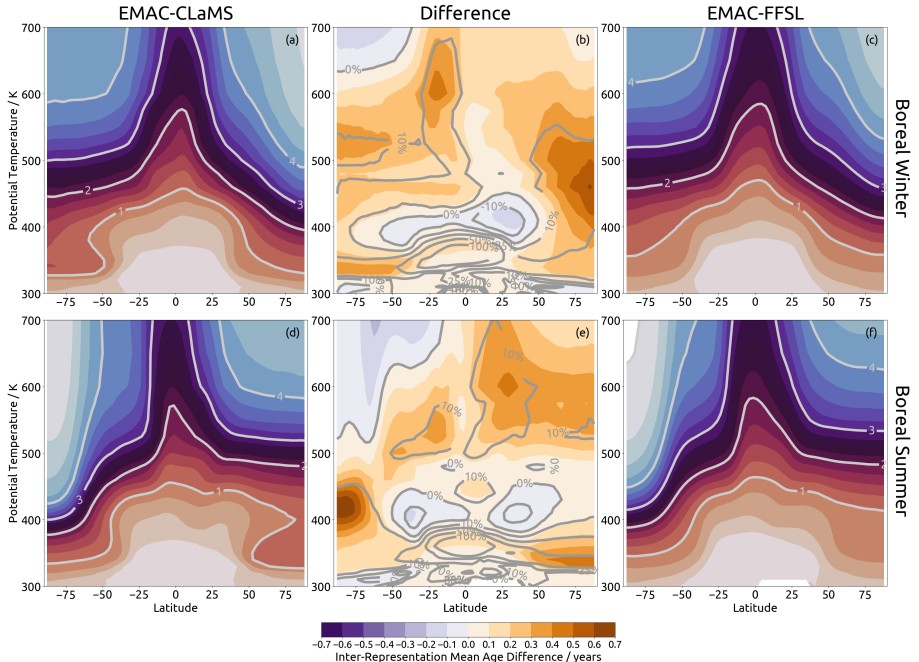

**Figure 1.** Mean age of air computed from age spectra for EMAC-CLaMS (a, d) and EMAC-FFSL (c, f) and the difference between them (b, e) in boreal winter (mean of December, January, and February) (a, b, c) and boreal summer (mean of June, July, and August) (d, e, f). For the central figures, shading shows the absolute differences (in years) between the representations (EMAC-CLaMS minus EMAC-FFSL) and contours show percentage differences (with EMAC-FFSL as baseline). Otherwise, contours and shading show mean age (in years), with a shading interval of 0.25 years.

intrusion of young air from lower latitudes). This structure in the mean age distribution agrees well with satellite observations (Stiller et al., 2012) and other models (e.g., Hauck et al., 2019).

The Lagrangian approach results in older air throughout most regions of the stratosphere. Above about 450 K, these differences are of quantitative nature and qualitatively the mean age distributions are similar during both seasons. A closer look shows that the particular contours are in somewhat different positions, especially around the polar vortexes. In particular, EMAC-CLaMS results show a deeper extent of old polar vortex air than EMAC-FFSL, most easily seen in the 3- and 4- year contours.

Below 450 K there are clear qualitative differences between the representations, most visible in the 1-year contour. This contour has nearly the same shape in the winter hemispheres in both transport schemes, but in the summer hemisphere this contour shows a qualitative inter-representation difference, particularly between 50 and 75 degrees latitude. In this region, between 350 K to 400 K, the contour shows an eave (a vertical inversion with young air extending over the subtropics, resembling a roof) in EMAC-CLaMS, but in EMAC-FFSL this contour rises towards the equator without showing an eave structure. In EMAC-CLaMS, the eave structure was found in the northern hemisphere during January in each year of the simulation, was



less pronounced during October, November, and February, and was not found in any month during any year in the EMAC-FFSL results. For the southern hemisphere, the eave structure was found in the EMAC-CLaMS results in July and was less pronounced during April, June, and August. The inter-representation mean age differences which are associated with this eave structure are approximately half a year.

Quantitative differences are largest within the polar vortexes, with higher mean age in EMAC-CLaMS. Other comparison studies of Lagrangian and Eulerian transport have already found that Lagrangian transport produces higher mean age within the polar vortexes due to stronger vortex edge transport barriers (Stenke et al., 2008; Hoppe et al., 2014). The results of this work echo those findings, and show a slightly stronger inter-representation discrepancy in the southern polar vortex, reaching a maximum of 0.7 years (compared to 0.6 years in the northern polar vortex). The southern polar vortex also shows stronger

confinement of the mean age differences, compared to the northern hemisphere. In particular, the 0.4 year contour around the southern polar vortex extends to 75° S, while in the north it extends nearly to 50° N. These results are likely due to the greater dynamical variability in the northern polar vortex (Butler et al., 2017). This greater dynamical variability likely causes blurring of the inter-representation discrepancy there, compared to the more consistent southern polar vortex.

Above 450 K, air is mostly older in EMAC-CLaMS than EMAC-FFSL. The largest differences occur at the edges of the

tropical pipe (around 25° N/S) and in the summertime middle and high latitude stratosphere. Above 500 K in southern high latitudes, EMAC-CLaMS shows younger air than EMAC-FFSL. These differences could be caused by recirculation differences, but are likely to be impacted by the differences in the upper boundaries of the two transport schemes (see Section 2.1) and will therefore not be investigated further.

There are several other regions with notable quantitative inter-representation differences in mean age. On the northern and

southern flanks of the region of horizontal outflow from the tropical tropopause layer (around 35° N/S and 400 K) EMAC-CLaMS shows younger air than EMAC-FFSL. This difference is stronger in the winter hemihere (greater than 0.5 years) and weaker in the summer hemisphere (less than 0.5 years). Although these differences are much weaker compared to the differences in the polar vortexes, they are rather large when the mean age in these regions is considered (approximately 50% of mean age, similar to the polar vortexes). The differences in these regions are the counterparts to those within the polar vortexes;

in the lower stratosphere EMAC-FFSL has older air near the boundaries of the tropical stratosphere and younger air within the polar vortexes due to stronger diffusion across the latitudinal age gradient along the polar vortex edge, creating a dipole feature in mean age differences.

## 3.2   Chemical Composition

Inter-reperentation differences in mean age are caused by differences in transport, meaning that simulations with chemically-

active tracers would also show corresponding differences in chemical composition. As an example, in Figure 2 we consider an idealized chemical tracer with a 2 year lifetime and an exponential decay globally (analogous to the E90 tracer commonly used to evaluate model transport, e.g., (Prather et al., 2011; Abalos et al., 2017), see Section 2.3 for details), which we assume to have been emitted from the tropical surface at a mixing ratio of 1 ppbv. Difference patterns in this 2 year lifetime tracer are largely a mirror image of differences in mean age, as larger age means greater chemical loss for the idealized tracer




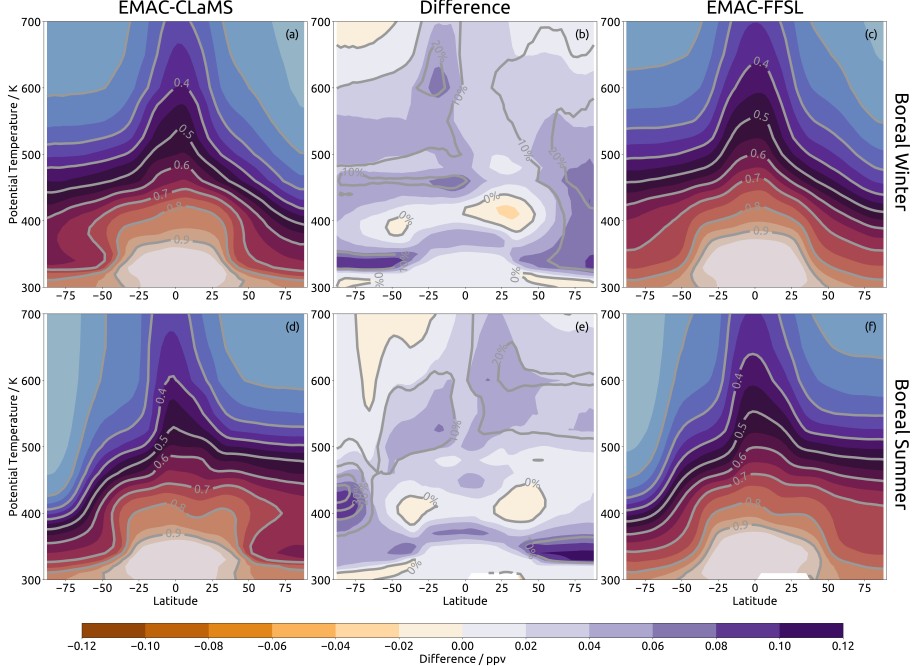

**Figure 2.** Same as Figure 1 but the quantity examined is a "forward-tracer" of 2-year lifetime (see text for details), with the exception that the percentage differences show in panels b and c use EMAC-CLaMS as the baseline (i.e. 30% means that EMAC-FFSL results shows 30% more forward tracer than those of EMAC-CLaMS).

from the original mixing ratio. However, the regions of highest sensitivity to the transport scheme differ somewhat for the 2 year tracer compared to mean age, as the tracer is less sensitive to changes in the spectrum tail. Maximum differences in tracer amount between EMAC-FFSL and EMAC-CLaMS are found in the polar vortex (up to 40%) and in the summertime lowermost stratosphere (up to 20%). These results suggest that there could be substantial impacts of the chosen transport

5   scheme on resulting chemical composition in these regions. Quantitative differences in the regions, however, depend on the tracer lifetime, and in the case of realistic observed chemical species, the particular sources and sinks of those species.

Figure 3 shows inter-representation differences in forward tracer mixing ratios at various locations for exponential decay lifetimes ranging from one tenth of a year to ten years. In all locations and for all lifetimes, EMAC-FFSL shows larger tracer mixing ratios than EMAC-CLaMS, related to younger age in these regions (compare Figure 1). The lifetime of highest

10   sensitivity to the transport scheme varies considerably between the different regions. In the lowermost stratosphere maximum differences occur for trace gas species with a lifetime of a few months (red lines). In the polar vortex, on the other hand, maximum differences occur for lifetimes of a few years (blues). Relative differences (in percent) show a different dependency on lifetime (monotonic decrease), as the tracer mixing ratio decreases with lifetime at a given location (Figure 3). For short lifetimes, relative differences grow enormously in some regions. For instance in the polar vortex (both NH and SH) EMAC-

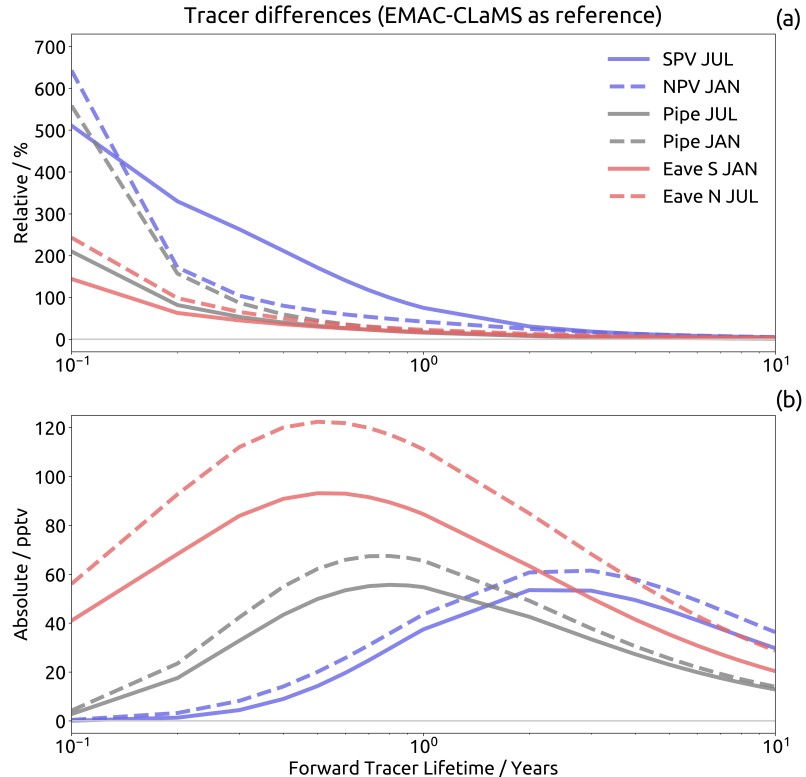

**Figure 3.** Inter-representation difference (a: absolute difference, EMAC-FFSL minus EMAC-CLaMS; b: relative difference, EMAC-CLaMS as reference) in forward tracer mixing ratios in several regions during January ("JAN") and July ("JUL"), versus exponential decay lifetime of the tracer. Results are shown for the southern polar vortex ("SPV", 70-90° S, 450 K), northern polar vortex ("NPV", 70-90° S, 480 K), tropical pipe ("Pipe", 5° S–5° N, 500 K), and summertime eave locations ("Eave", 50°−75° north or south, 360 K).

FFSL tracer mixing ratios are higher than for EMAC-CLaMS by up to a factor 5. The southern polar vortex stands out as a region with extremely large differences in the entire lifetime range below about 2 years.

To investigate the relation between tracer differences and the representation of stratospheric barriers in the results of the two transport schemes, Figure 4 presents horizontal and vertical gradients of the 2-year lifetime forward tracer. Broadly speaking, the vertical gradients are strongest along the tropopause, while the horizontal gradients are strongest at the subtropical jets, the polar vortexes (most strongly at the southern polar vortex), and the edges of the tropical pipe. While this is true in the results of both transport schemes, EMAC-CLaMS always shows gradients which are as strong or stronger than those of EMAC-FFSL. In particular, the vertical gradients at the extratropical tropopause are approximately twice as strong in EMAC-CLaMS, as are the horizontal gradients at the southern polar vortex and the edges of the tropical pipe. Meanwhile the horizontal gradients



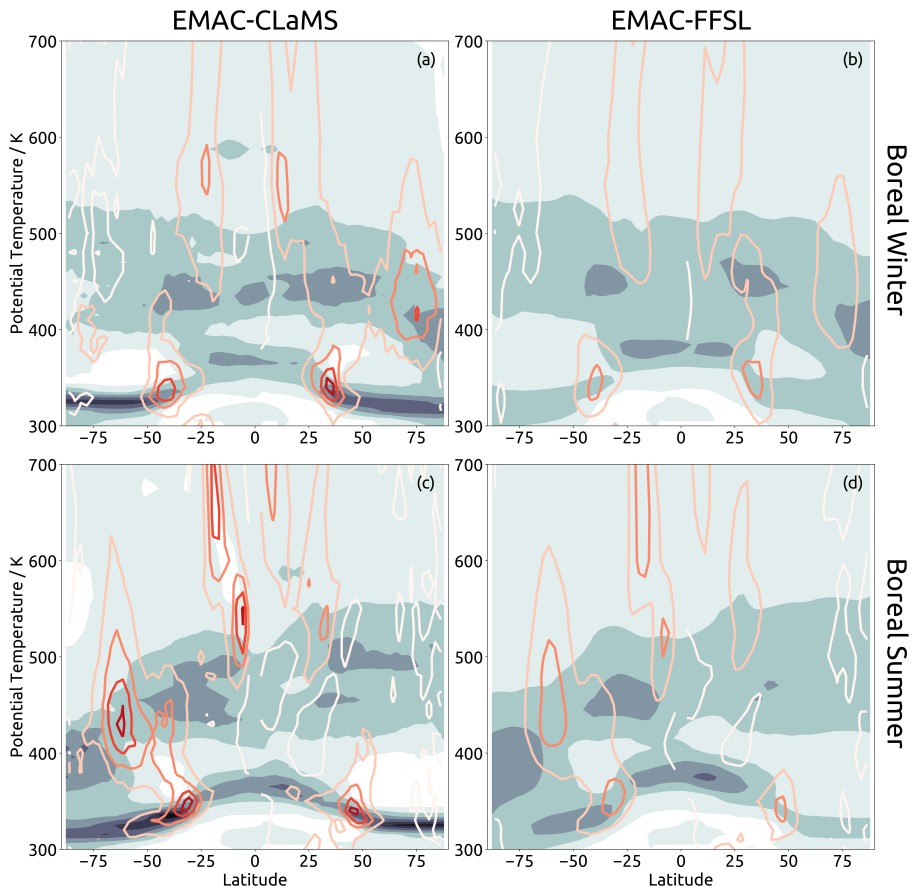

**Figure 4.** Gradients of a 2-year lifetime forward tracer, from the tracer field calculated by the 10-year mean of representation age spectra. Shown are results from EMAC-CLaMS (a, c) and EMAC-FFSL (b, d) during January (a, b) and July (c, d). The vertical gradient is calculated with respect to potential temperature and shown in the grey shading while the horizontal gradient is calculated with respect to the absolute value of latitude and shown with the colored line contours. Plotted gradients do not have explicit units; the vertical (horizontal) gradient is normalized so that the darkest (reddest) shading (contour) corresponds to the maximum vertical (horizontal) gradient found in all four panels, with linear steps from 0 to the maximum value for each contour line.

at the subtropical jets are approximately 50% stronger in EMAC-CLaMS than in EMAC-FFSL. These results suggest that the representation of transport barriers is substantially stronger in EMAC-CLaMS than in EMAC-FFSL. While this has been shown for the case of the polar vortex already by Hoppe et al. (2014) the analysis here generalizes these findings to all the aforementioned stratospheric transport barriers.

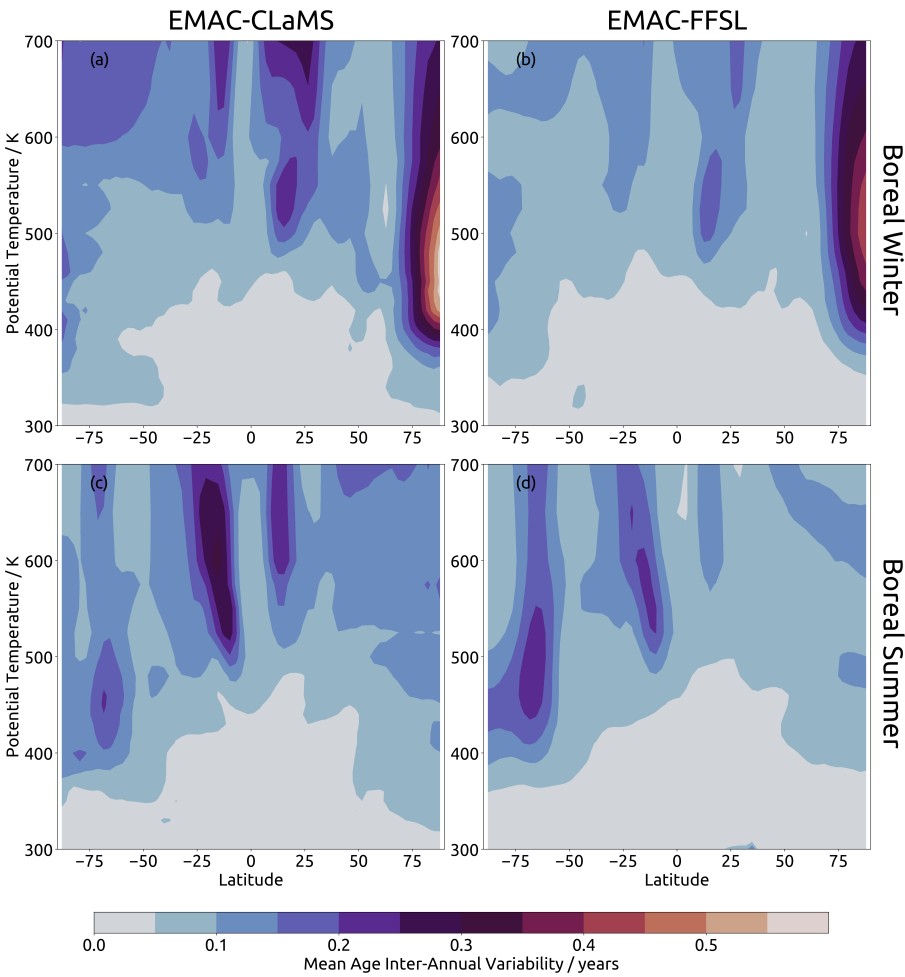

**Figure 5.** Sandard devation of spectra monthly-average mean age over the ten-year climatology from EMAC-CLaMS (a, c) and EMAC-FFSL (b, d) during boreal winter (a, b) and summer (c, d).

## 3.3 Inter-annual variability

Inter-annual variability in the mean age fields is shown in Figure 5. The results show clear inter-representation differences, indicating that simulated inter-annual transport variability is affected by the choice of transport scheme. In both representations the greatest variability is found in the northern polar vortex and second to that at the edges of the tropical pipe. Whereas high mean age variability is found in the center of the northern polar vortex, for the southern polar vortex the strongest mean age variability is found at the edge of the vortex. This is the case in both schemes, and is likely to be primarily related to the frequency of sudden stratospheric warmings, which occur much more often in the northern polar vortex than the southern polar vortex. In EMAC-FFSL, the mean age variability at the southern polar vortex edge is roughly equal to the variability found at the edges of the tropical pipe. However, in EMAC-CLaMS the variability at the edges of the tropical pipe is roughly twice





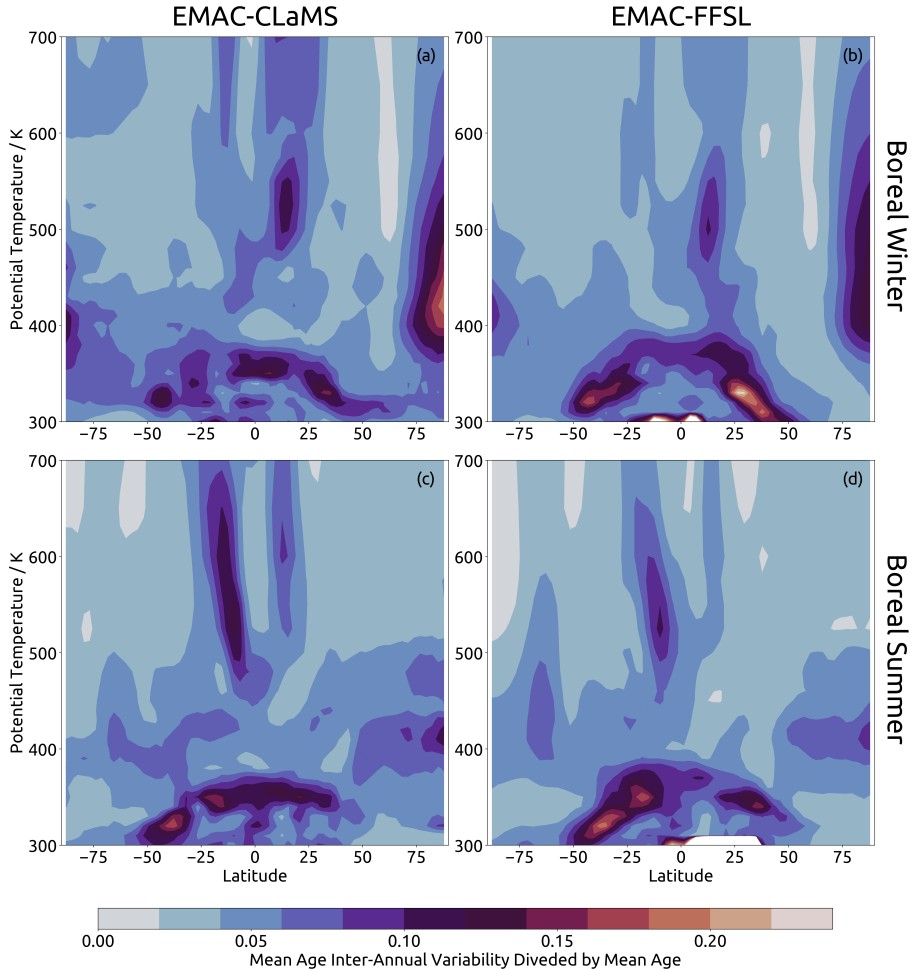

**Figure 6.** Same as Figure 5, but the quantity shown is the standard deviation of spectra mean age scaled (divided) by the spectra mean age.

as strong as the variability at the edge of the southern polar vortex. The inter-representation difference in this comparison is partially due to stronger southern polar vortex edge variability in EMAC-FFSL than EMAC-CLaMS. However, this discrepancy is smaller than the inter-representation difference in tropical pipe edge variability; variability at the tropical pipe edges is about twice as strong in EMAC-CLaMS as in EMAC-FFSL. This is also the case in the northern polar vortex, where mean age

5    variability is about 50% stronger in EMAC-CLaMS than in EMAC-FFSL.

Figure 6 shows inter-annual variability normalized by local mean age. From this prespective, the northern polar vortex still appears as a hotspot of variability and is still stronger in EMAC-CLaMS than in EMAC-FFSL. Conversely, the southern polar vortex edge shows much weaker variability compared to other locations, due to high mean age values in that region, and appears to have variability of approximately equal magnitude in both representations. The largest difference in this perspective from

10   that of absolute difference values is found around the tropical tropopause. Variability in this location is stronger in EMAC-FFSL





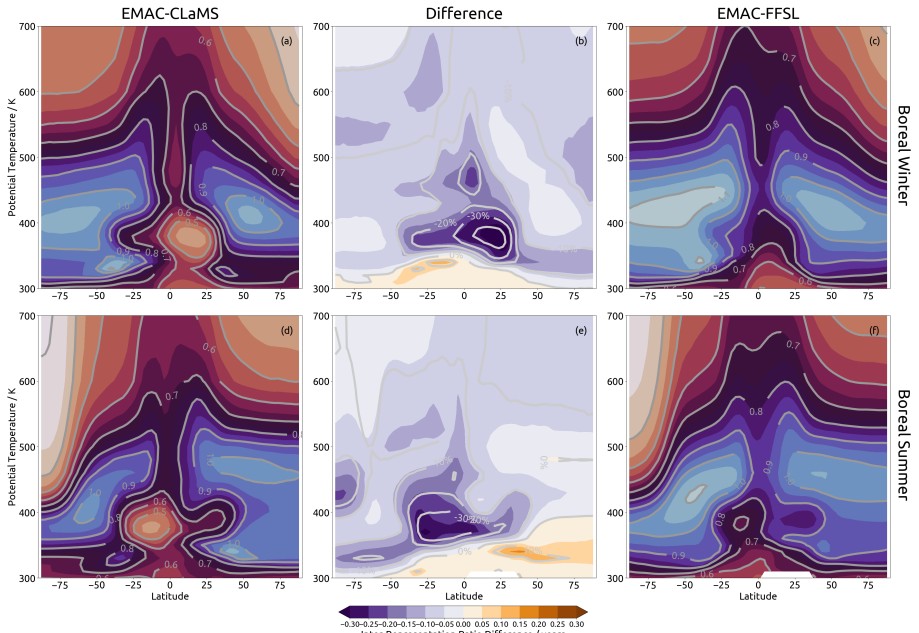

**Figure 7.** Panels correspond to those of Figure 5, but the quantity shown in the age spectra ratio of moments (width divided by mean age, units of years).

than in EMAC-CLaMS. Furthermore, in EMAC-FFSL this variability is strongest beyond the subtropical jets, rather than at the tropical tropopause (i.e. equatorward and upward of the subtropical jets). In the case of EMAC-CLaMS, variability beyond the subtropical jets is of a similar magnitude to variability along the tropical tropopause. These findings could indicate a critical role for transport across the subtropical jets to cause the differences in the eave structures in the age distribution between the

5 Lagrangian and Eulerian frameworks (see Figure 1). Analysis of the age spectra in Section 4.3 will shed more light on the reasons for the occurrence of the eaves.

### 3.4 Age spectrum shape

The age spectra width is defined as the second moment of the spectra centered around the mean (e.g., Waugh and Hall, 2002)

$$\Delta^2 = \frac{1}{2} \int\limits_0^\infty (\tau - \Gamma)^2 \, G(\boldsymbol{r}, t, \tau) \, \mathrm{d}\tau \, . \tag{4}$$

10 The width quantifies the spread or dispersion of the spectra.

Spectra width ranges from near zero to almost 2.5, with the lowest values found in the troposphere and the highest values found in the most troposphere-remote regions of the stratosphere, like the extratropical middle stratosphere and the polar vortexes (not shown). The summertime eave pattern in EMAC-CLaMS found in mean age and forward tracer contours is also seen in spectra width as a region of higher widths (not shown).





An important parameter characterizing the shape of the age spectra is the "ratio of moments", the spectra width divided by the mean $\Delta^2/\Gamma$. The ratio of moments is also a critical parameter for estimating mean age from trace gas measurements (e.g., Volk et al., 1997; Bönisch et al., 2009; Engel et al., 2009).

Figure 7 shows the ratio of moments from the age spectra. In general, the ratio of moments is relatively small in the tropics,
related to narrow age spectra there, and increases in middle latitudes where age spectra are broader. The ratio of moments is larger in the summer compared to the winter hemisphere. The decrease at the upper levels and in the polar vortex is, to some degree, related to the truncation of the spectra at 10 years, which causes some underestimate of age spectra width. The patterns agree qualitatively with results from other models (e.g., Hall and Plumb, 1994; Hauck et al., 2019). Quantitatively, the ratio values are lower than those found in the recent study by (Hauck et al., 2019), which is related to the truncation of the spectrum
tail here and should not be viewed as contrary to those results.

The inter-representation ratio differences (Figure 7, b and e) show that the ratio of moments (hence the spectrum shape) is sensitive to the transport scheme used. Throughout most regions of the stratosphere, the ratio of moments is larger in EMAC-FFSL than EMAC-CLaMS. The largest differences (up to 40%) occur in the winter hemisphere subtropics at potential temperature levels between about 350 and 450 K. In this location, EMAC-CLaMS shows a very localized region of low
spectrum moment ratios, while EMAC-FFSL shows a much weaker minima and only shows this in the southern tropics.

The summertime lowermost stratosphere is the only region where the ratio of moments is larger in EMAC-CLaMS than EMAC-FFSL. A remarkable feature is the vertical dipole in the summertime subtropical lowest stratosphere with larger ratios below (around 350 K) smaller ratios (around 380 K). In other words, at this location relatively broad spectra reside below narrower spectra. This characteristic in the ratio of moments is much more clear in EMAC-CLaMS than in EMAC-FFSL and
is likely related to the eave structures found in the mean age distribution in EMAC-CLaMS. The details of the age spectra in this region will be investigated in Section 4.3.

## 4  Differences in the representation of transport processes

To gain further insight into inter-representation differences in transport processes, we turn our investigation to the stratospheric age spectrum. This section is subdivided according to the regions with the most significant differences: the tropical and mid-
latitude stratosphere, the polar vortexes, and the lowermost stratosphere.

### 4.1  Tropical and mid-latitude stratosphere

Air enters the stratosphere across the tropical tropopause in the TTL and is then transported upwards in the tropical pipe or poleward by the shallow branch of the BDC. Within the tropical pipe, with its lower edge at about 450 K, exchange with middle latitudes is suppressed and air is thereby largely confined therein (Plumb, 1996).
Figure 8 (a) shows age spectra from EMAC-FFSL and EMAC-CLaMS at the 500 K level during boreal winter (January) in the tropical pipe. Results during boreal summer are very similar (not shown). There is a clear shift of the EMAC-FFSL spectrum (red) towards younger ages for the transit time range below about 2 years, compared to EMAC-CLaMS. This shift

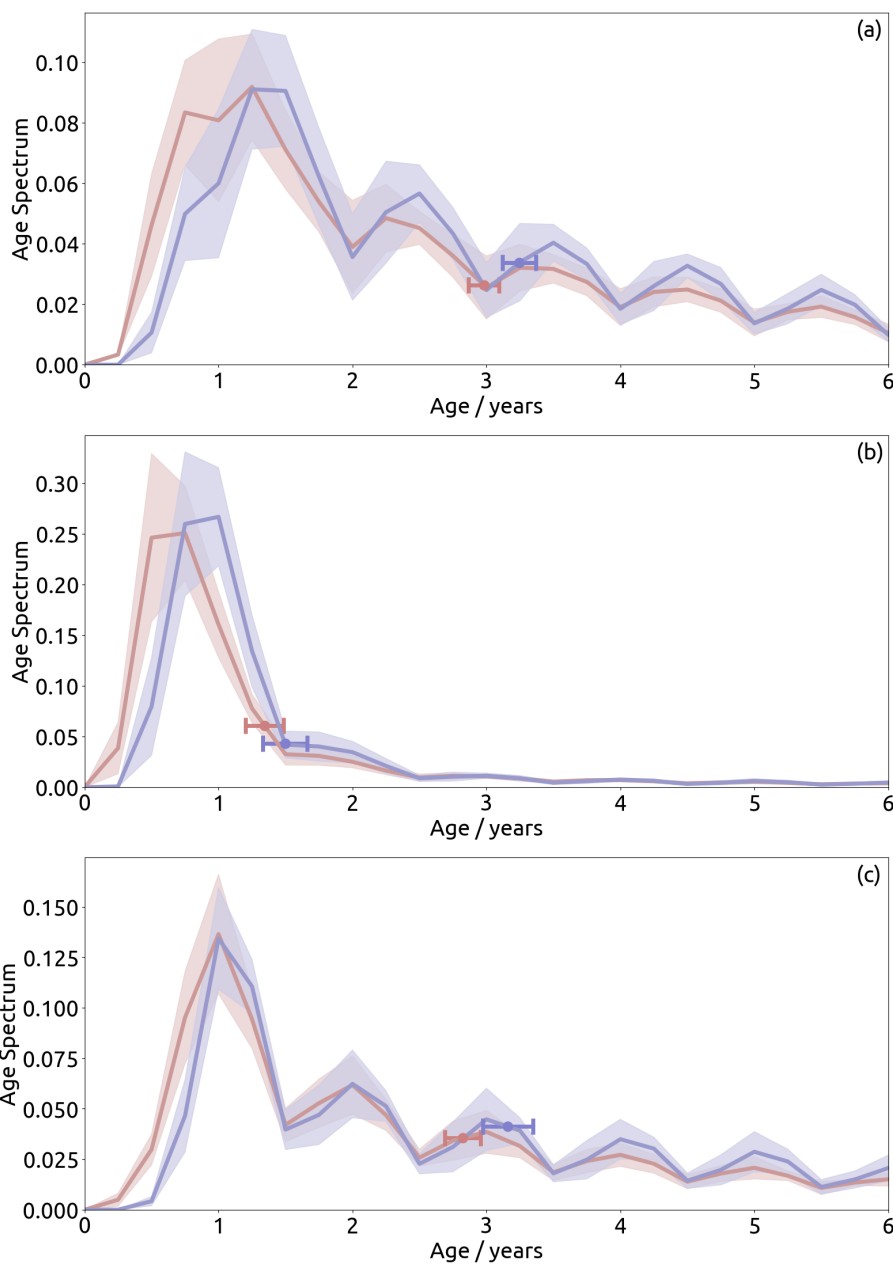

**Figure 8.** Age spectra from the results of EMAC-FFSL (red) and EMAC-CLaMS (blue) at 500 K for (a) the southern mid-latitude stratosphere (40-60° S, July), (b) the tropical pipe (6° S–6° N, January), and (c) the northern mid-latitude stratosphere (40-60° N, January). Lines indicate multi-annual mean with shading showing annual variability. Dots indicate mean age of spectra, with surrounding bars showing annual variability. Variability for both quantities is computed as two standard deviations.





shows a higher fraction of air younger than 9 months in EMAC-FFSL, resulting from much faster tropical upward transport from that transport scheme. For a transit time of 3 months (which is the age spectrum resolution, see Section 2) EMAC-FFSL shows a substantial air mass fraction of about 4 percent whereas in EMAC-CLaMS there is no such air. Results for the next transit time bin (at 6 months) are similar: the EMAC-FFSL air mass fraction is significantly larger than for EMAC-CLaMS

(about 0.25 compared to 0.075). As 3 months is beyond the fastest transit time from the middle troposphere to 500 K based on large-scale upwelling velocities (Wright and Fueglistaler, 2013), the differences at short transit times can only be caused by stronger vertical diffusion due to numerical diffusion in the FFSL transport scheme.

Comparison of age spectra in Northern middle latitudes at the same level (Figure 8c) shows smaller differences and even the same modal age (defined as the transit time of the age spectrum peak) for EMAC-FFSL and EMAC-CLaMS. However, in

this case EMAC-FFSL transport again clearly shows a larger fraction of young air with transit times less than a year. Similar to the case of tropical transport these differences must be related to stronger numerical diffusion in the EMAC-FFSL transport scheme. Another interesting feature is the stronger multiple peaks in the spectrum tail for EMAC-CLaMS (from ages of 3 years above). The occurrence of multiple peaks in stratospheric age spectra is caused by the seasonality of transport into the stratosphere (Reithmeier and Sausen, 2008; Ploeger and Birner, 2016). Stronger numerical diffusion in EMAC-FFSL blurs this

seasonal transport signal over the course of a few years.

Very similar conclusions hold for the southern hemisphere middle latitudes during austral winter (Figure 8a); The fraction of young air (age below about 1 year) here is greater in EMAC-FFSL compared to EMAC-CLaMS, related to stronger diffusion of the transport scheme, and the peaks in the spectrum tail are again weaker.

## 4.2 Polar vortex

Due to strong polar downwelling motion and the cyclonic circumpolar flow, air masses inside the wintertime stratospheric polar vortex are largely isolated against exchange with middle latitudes. Figure 9 shows the age spectra within the southern stratospheric polar vortex. Below 3 years, the spectra show clear qualitative differences. EMAC-FFSL shows two peaks in this region: one at 2.5 years and the other at 1.25 years. Meanwhile EMAC-CLaMS shows only one peak, which is at 2.5 years. The common peak at 2.5 years is much stronger in EMAC-CLaMS than in EMAC-FFSL. The contribution from air younger

than 2 years is about twice as strong in EMAC-FFSL as in EMAC-CLaMS, and at ages less than 1 year this difference is even stronger. This much higher fraction of young air inside the polar vortex in EMAC-FFSL than in EMAC-CLaMS is caused by stronger diffusive transport across the vortex edge in the FFSL transport scheme. This difference suggests that simulations of chemically-active tracers with short stratospheric lifetimes and tropospheric origins would show substantially stronger southern polar vortex concentrations in EMAC-FFSL, compared to EMAC-CLaMS. For long-lived trace gas species differences would

be smaller. Consequently, the amount of ozone depleting substances in polar regions with lifetimes below a few years and related polar ozone loss can substantially differ depending on the chosen transport scheme.

As a side note, variability in the age spectra seem to be roughly similar at most ages, but is substantially different below 3 years of age, with much more variability in EMAC-CLaMS at the 2.5 years peak and much more variability in EMAC-FFSL below 2 years of age. At ages older than 3 years the age spectra are qualitatively similar, showing multiple maxima at 1-year



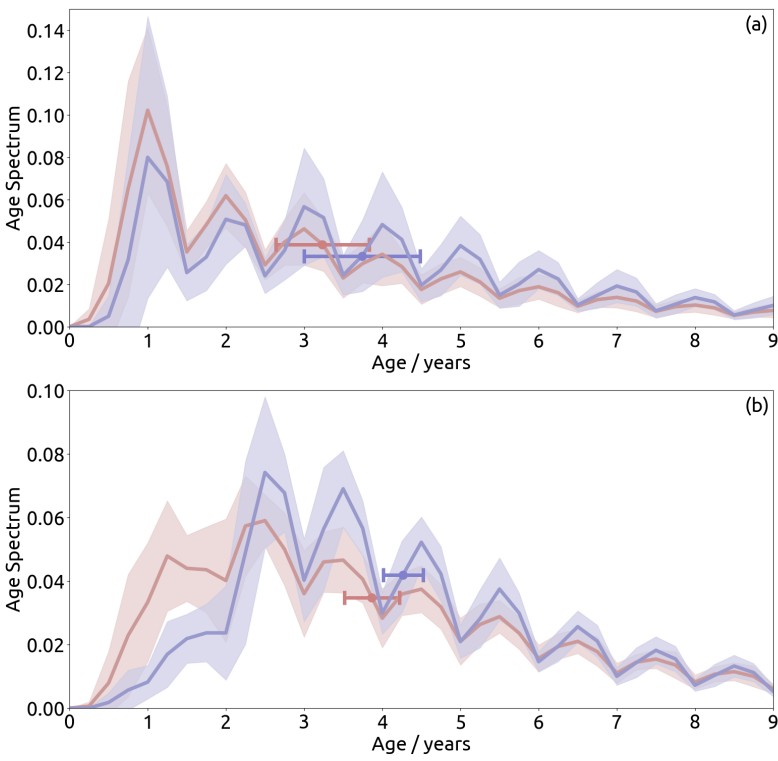

**Figure 9.** Age spectra from EMAC-FFSL (red) and EMAC-CLaMS (blue) within (a) the northern polar vortex (480 K, 70–90° N, January).
and (b) the southern polar vortex (90-70° S 450 K, July), Lines indicate multi-annual mean with shading showing annual variability. Dots
indicate mean age of spectra, with surrounding bars showing annual variability. Variability for both quantities is computed as two standard
deviations.

intervals at the half-year marks, and regular minima at the 1-year marks. This means stronger contribution of air emitted during
January, and weaker contribution of air emitted during July. Both schemes show this quality, with EMAC-CLaMS showing a
greater difference between the contributions at the maxima and minima.

Figure 9 shows age spectra within the northern polar vortex. As in the southern polar vortex, ages above 3 years show
5   qualitative similarity between the representations; maxima in the spectra correspond to January-emitted tracers while minima
correspond to July-emitted tracers. At ages younger than 2.75 years, EMAC-FFSL shows greater tracer concentrations than
EMAC-CLaMS. However, the difference between the two representationss in this location for young ages is much smaller than
the difference in the southern polar vortex, while variability in the age spectra is much stronger (approximately a factor of 2)
in EMAC-CLaMS than in EMAC-FFSL.

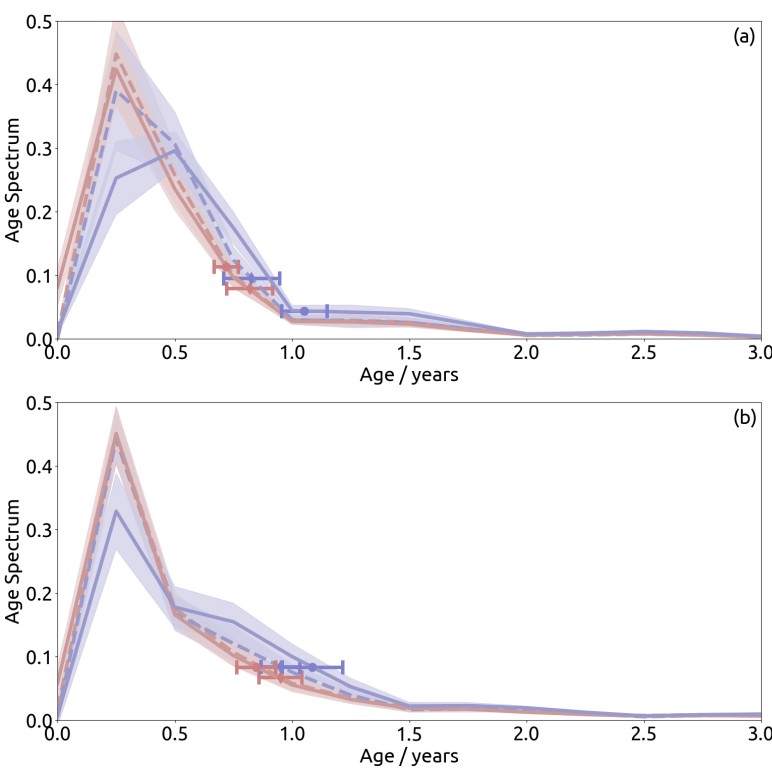

**Figure 10.** Age spectra from the results of EMAC-FFSL (red) and EMAC-CLaMS (blue) within the summertime lowermost stratosphere regions of the eave structures at 360 K (sold lines) and 400 K (dashed lines) in (a) Southern hemisphere (55–75° S, January), and (b) Northern hemisphere (55–75° N, July).

## 4.3 Lowermost Stratosphere

A particularly interesting feature in the mean age and tracer distributions in the summertime lowermost stratosphere in Figures 1 and 2 is the eave structure in EMAC-CLaMS, an age inversion with a layer of younger air residing above older air, which is totally absent in EMAC-FFSL. The structure seen in EMAC-CLaMS has two features: an old-air region at the level of the subtropical jet (around 350 K) and a young-air region above that (around 400 K). Conversely, the EMAC-FFSL lower region has an age which is similar to that in the upper region, and therefore the presence/absence of the eave structure depends on the characteristics of transport into the lower region. As the mean age and forward tracer contours in Figures 1 and 2 in the upper region follow similar paths in both representations, transport from the upper region into the lower region is not likely to play





a role in the discrepancy of the eave structure representation. Therefore, the eave structure seen in EMAC-CLaMS probably arises from weaker transport from the troposphere into the lower eave region, in comparison to EMAC-FFSL.

To gain more insight into the underlying processes, Figure 10 shows the corresponding age spectra for the two schemes at the 350 K and 400 K levels between 50-60 degrees latitude. In both cases, the upper level age spectra are very similar in

both EMAC-FFSL and EMAC-CLaMS. In the southern hemisphere in particular, these spectra are nearly identical, with only slightly more tracer between 0.5 and 1.5 years of age found in EMAC-CLaMS. Meanwhile the northern hemisphere results show somewhat less agreement between the two representations in the upper levels, with slightly less tracer at 0.25 years and somewhat more tracer between 0.5 and 1.0 years in EMAC-CLaMS. However, there is considerable inter-representation difference in the relationship between the age spectra in the upper region and the lower region; EMAC-FFSL results show

nearly identical spectra in both regions, while EMAC-CLaMS shows a consistent difference in the upper and lower region spectra. In the EMAC-CLaMS spectra for both hemispheres, the upper region shows more air younger than 0.5 years while the lower region shows more air between 0.5 years and 1.5 years, and both regions show nearly identical contributions from air at 0.5 years. The differences in age spectra, mean age and tracer mixing ratios suggest that the eave structure in the lowermost stratosphere is caused by an interplay of transport processes as described in the following.

The lowermost stratosphere age background results from a mixture of old air masses downwelling from the stratosphere, and young air masses transported into the region by the shallow branch of the BDC (e.g., Bönisch et al., 2009). In spring and summer, a new transport pathway emerges which is related to upward transport in the tropics and poleward transport directly above the subtropical jet, and characterized by transport time scales of about half a year to 1.5 years. This poleward transport happens in the layer of about 380–450 K, which belongs to the region above the jet and below the tropical pipe. Fast transport

in this layer agrees well with the existence of a tropically controlled transition region for water vapour as proposed by Rosenlof et al. (1997). The EMAC-CLaMS simulation shows a clear age inversion related to this flushing of the extratropical lowermost stratosphere with young air above the jet. In the EMAC-FFSL simulation, on the other hand, this feature is totally absent because a much higher fraction of young air with transit times shorter than 0.5 years blurs the old air signature in the layer around 350 K.

Hence, the Lagrangian and Eulerian transport schemes result in different preferences for transport pathways into the summertime lowermost stratosphere: poleward transport above the jet (Lagrangian) versus cross-tropopause transport at levels below (Eulerian). It remains to be shown from trace gas observations in the lowermost stratosphere whether the eave structure evident in the age distribution from Lagrangian transport is a feature of the real atmosphere. Initial indications for a mixture of old wintertime air and young air masses from transport above the subtropical jet in that region during early spring have already

been found in aircraft in-situ measurements of $N_2O$ and CO by Krause et al. (2018).

## 5   Discussion

The results of the work presented thus far have shown substantial differences in tracer transport between EMAC-FFSL and EMAC-CLaMS. Given that the FFSL transport scheme used by EMAC is also used in a wide array of other climate models, the





effects of unphysical numerical diffusion in EMAC-FFSL which have been described here are likely to affect tracer transport in other climate models as well. This could cause complications for the interpretation of results from these models, when the topic of interest is stratospheric transport. One such topic, for which there is considerable modeling activity at the moment, is geoengineering through stratospheric aerosol injection (SAI). This has been proposed as a method to reduce or entirely offset

the surface temperature effects of global warming (e.g., Crutzen, 2006) and is likely to gather more attention as the global mixing ratios of greenhouse gases rise. Relatedly, the latest-generation climate models from the Coupled Model Intercomparison Project phase 6 (CMIP6) show an even stronger equilibrium climate sensitivity and simulate stronger climate warming than the model generation before (Forster et al., 2020) further fueling discussion about solar geoengineering.

A modelling effort to assess the opportunities and risks of solar geoengineering using stratospheric sulfate aerosols within

the Geoengineering Large Ensemble (GLENS) project has recently been presented by (Tilmes et al., 2018). In this project, injection strategies have been proposed to maintain the distribution of global surface temperatures in the future and potential side-effects (e.g., on precipitation and stratospheric ozone) have been discussed (Kravitz and Douglas, 2020). Although the results of that work suggest that it may be possible to use SAI successfully (i.e., to maintain the global distribution of surface temperatures), the authors note that a main uncertainty in their model results is related to stratospheric transport processes and

their representation in current climate models.

Our model experiment, which applies one climate model with two different transport schemes in the same simulation, is well-suited to shed further light on this uncertainty of geoengineering projections related to uncertainties in air mass dispersal due to the model representation of stratospheric transport. It is noteworthy here that this discussion concerns air mass transport and not the transport of sulfate, as our simulation does not include stratospheric chemistry. However, we consider a state-of-

the-art transport scheme (EMAC-FFSL) which is also applied in other current climate models and a novel Lagrangian scheme (EMAC-CLaMS) which has significantly less numerical diffusion. As results from this paper show, two regions emerge where transport differences between the two representations are especially large: the lowermost stratosphere and the polar vortex. Both are critical regions for the processes which affect the efficacy of SAI. In particular, sulfate concentrations in the lowermost stratosphere crucially affect radiative forcing, whereas sulfate concentrations in the polar vortex control the side-effects of

geoengineering on stratospheric ozone.

To illustrate the potential differences in geoengineering simulations caused by model transport representation, we modified our experiments to include continous point-source injections of tracers with idealized chemistry. The injection is handled by forcing the tracer mixing ratio to 1 ppbv within a region of nine EMAC grid cells (3-cells wide both east-west and north-south). The idealized chemistry is represented by a global exponential decrease with 30-, 90-, and 365-day lifetimes. Figure 11 shows

the dispersal of a 365-day lifetime tracer which was injected at 30° N and 180° E at the 89 hPa pressure level. The results are shown for the two transport schemes after about 5 years of simulation and the results represent the state of the simulation on a single timestep. Both models show three regions with high tracer mixing ratios: (1) a plume between 300° E and 330° E which is the most prominent feature of the snapshot; (2) a second plume west of 260E and between 40-50S; (3) and then a third local maxima of tracer mixing ratios in the upper northwest corner of the image. In the EMAC-FFSL results this latter region seems

to be separate from the others in the image, while in EMAC-CLaMS this region seems to be connected to the main plume by a





trail of weaker tracer mixing ratios. In both features (1) and (2), EMAC-CLaMS results show higher mixing ratios in the centers of the plumes. In feature (1), these mixing ratios even reach nearly as high as the emission mixing ratio (1 ppbv), showing that the central area of the plume remained isolated during transport over 60 degrees of longitude. In comparison, the highest mixing ratios found in EMAC-FFSL are about 0.45 ppbv - half the emission mixing ratio. Furthermore, there is clearly a much wider variety of small-scale features in the results of EMAC-CLaMS compared to those from EMAC. Hence, the stronger numerical diffusion in EMAC's FFSL transport scheme blurs small-scale features and filaments compared to Lagrangian transport and results in a more homogeneous tracer distribution.

Global tracer distributions from the two models at the end of the 5 year simulation period (for the 365 days lifetime tracer) are shown in Figure 12 for the case of austral spring (September–November). The tracer plume extending from the injection source location in the southern subtropics towards the south pole is broader and more smeared out in EMAC-FFSL than EMAC-CLaMS, also related to the differences in numerical diffusion. The difference figure (Figure 12) indicates even clearer that for EMAC-CLaMS the plume is more centered around its core whereas for EMAC-FFSL it is broader with more tracer above and below. In particular inside the polar vortex (poleward of about 60° S), tracer mixing ratios are substantially (approximately 35%) higher for the more diffusive FFSL transport scheme.

These differences emerge for all injected tracers considered, including over each of the lifetimes of 30, 90, and 365 days. We therefore expect that for realistic chemistry there should also be significantly higher sulfur concentrations in polar regions for more diffusive model transport schemes, compared to Lagrangian schemes. As relative differences in the polar vortex are substantial, we expect a large uncertainty of simulated ozone depletion from geoengineering sulfur injections related to the used model transport scheme. Narrowing this uncertainty further down, in particular using simulations including appropriate stratospheric chemistry for sulfur and ozone, should be a priority for future research in this direction. For the moment, in view of such large uncertainties in stratospheric transport in current models and the potential dangers of SAI geoengineering, real-world applications of SAI remain highly questionable and inadvisable.

## 6 Conclusions

In this work, we have assessed the impact of the choice of trace gas transport scheme on the representation of stratospheric transport. The two transport schemes that we have studied are the Lagrangian scheme of CLaMS and the Eulerian FFSL scheme of EMAC, the latter of which is commonly used in modern chemistry-climate models. Differences in transport time scales were investigated by comparing the full time-dependent age spectrum and idealized, radioactively-decaying forward tracers in representations from both schemes. The results show that stratospheric transport barriers are, in general, much stronger in simulations with Lagrangian trace gas transport whereas they are weaker for the FFSL scheme due to stronger, unphysical numerical diffusion associated with the latter method. These conclusions hold for the transport barriers around the polar vortex, along the subtropical jets, and at the edges of the tropical pipe. Two regions of the stratosphere emerge from the simulations for which differences caused by the transport scheme are particularly large: (i) the polar vortex and (ii) the summertime lowermost stratosphere. Inside the polar vortex, the air is substantially older in the Lagrangian transport simulation due to

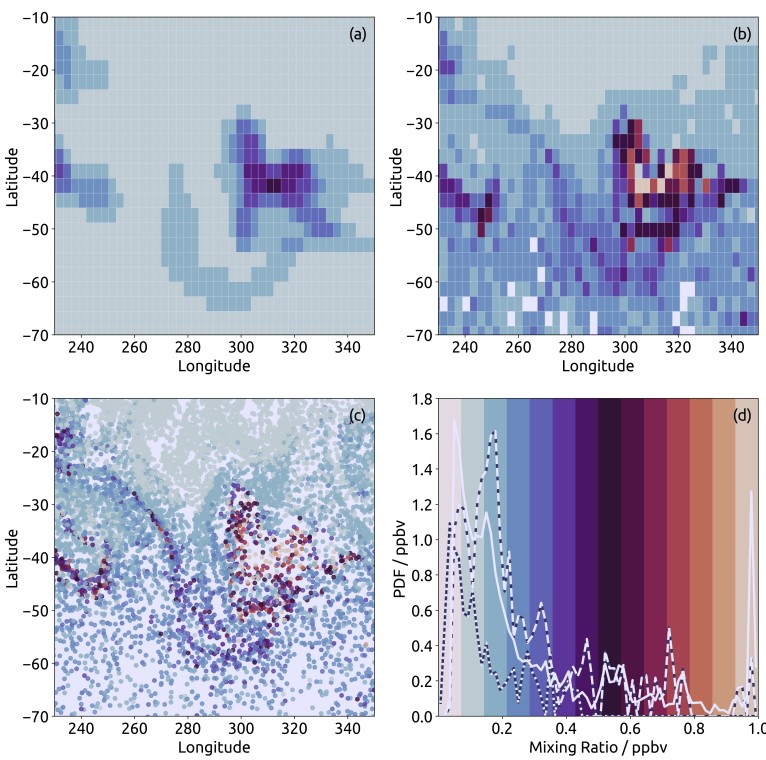

**Figure 11.** Region from plume injection experiment showing results for a long-lived (365-day lifetime) tracer. Panel (a): EMAC-FFSL results on the EMAC grid at model level 63 (approximately 100 hPa, and the level at which plume injection occured). Panel (b): EMAC-CLaMS results gridded onto the EMAC model grid at the same level as the EMAC results. Panel (c): EMAC-CLaMS data for parcels within EMAC model level 63 in the unprocessed Lagrangian representation. Panel (d): histograms showing distributions of tracer mixing ratios within the shown region. The color map used in panels a-c corresponds to the background colors in panel d. Histograms are shown for unprocessed EMAC-CLaMS results (solid line, corresponding to distribution in panel c), EMAC-CLaMS gridded results (long-dashed line, corresponding to distribution in panel b), and EMAC-FFSL results (short-dashed line, corresponding to distribution in panel a). Histograms are computed using only data which is shown in the other three panels (i.e., within the shown region and within EMAC model level 63), and the histograms of gridded results are mass-weighted.

reduced diffusive transport from middle latitudes through the vortex edge. Consequently, chemical tracers with short lifetimes show much lower mixing ratios. Also in the lowermost stratosphere, the air is much older for the Lagrangian simulation, as diffusive cross-tropopause transport of young air from the troposphere is reduced.



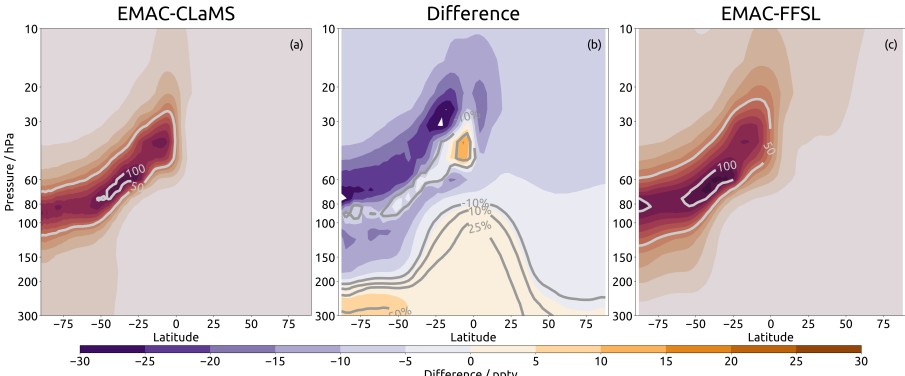

**Figure 12.** Zonal mean tracer distribution from continuous mass injection in the stratosphere (30S, 100hPa) from EMAC-CLaMS (a) and EMAC-FFSL (c) results, contours showing tracer mixing ratios in pptv (emission value of 1 ppbv). Also show is the difference between the fields (b) with both absolute differences (shading) and percentage differences (contours, EMAC-CLaMS as reference). Results are shown for a 1-year lifetime tracer.

In particular, a very different structure in the age of air and tracer distributions emerges in the summertime lowermost stratosphere in the two representations. The Lagrangian representation of EMAC-CLaMS shows an age inversion structure, or eave, where older air resides below younger air, while this feature is entirely absent in the EMAC-FFSL results. This structure is related to fast poleward transport above the jet, which creates the young air layer above the older air. In the EMAC-FFSL results, strong diffusive cross-tropopause transport totally blurs this layered structure.

The results of this paper show that a fully Lagrangian transport scheme (that of CLaMS) results in significantly less numerical diffusion, stronger stratospheric transport barriers, and clearer structures in trace gas distributions (e.g., gradients, filaments), even when compared to a sophisticated, state-of-the-art flux-form semi-Lagrangian scheme (that of EMAC). Differences in simulated trace gas transport related to the choice of the transport scheme raise important questions about the uncertainty of stratospheric transport in climate model simulation, and in particular for geoengineering model experiments.

*Author contributions.* Edward Charlesworth and Felix Plöger prepared the manuscript of this work with assistance from all other authors. Model development was performed by Edward Charlesworth, Frauke Fritsch, and Patrick Jöckel. Simulations were executed by Edward Charlesworth and Ann-Kristin Dugstad. All authors contributed to analysis and interpretation of results.

*Competing interests.* The authors declare no competing interests.



*Acknowledgements.* We would like to thank Nicole Thomas for her invaluable technical assistance as well as Hella Garny, Marius Hauck, Peter Hoor, with whom we had very stimulating discussions. This work was funded by the Helmholtz Association under grant no. VH-NG-1128 (Helmholtz Young Investigators Group A-SPECi), and was also enabled by the Helmholtz Association Earth System Modeling project Finally, we gratefully acknowledge the computing time for the EMAC simulations which was granted on the supercomputer JURECA at the

5  Jülich Supercomputing Centre (JSC) under the VSR project ID JICG11.



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
