# Peer review of "Impact of Lagrangian Transport on Lower-Stratospheric Transport Time Scales in a Climate Model"

_Atmospheric Chemistry and Physics, 2020_

## Referee Comment (RC1) · Anonymous Referee #1 · 7 Aug 2020

**Response**

Accurate representation of transport in climate models is difficult. Unwanted numerical diffusion introduces artifical fluxes in the stratosphere which prevent representation of tracer gradients with fidelity. This issue is more prominent in the stratosphere where three key transport barriers - one in the subtropics, the other at the vortex edge and the third near the tropopause - exist. Artifical diffusive fluxes tend to enhance transport across these barriers inducing unwanted mixing of tracers.

Even though such fluxes should ideally vanish at high grid resolutions, integrating comprehensive climate models at such resolutions is not practically possible. Therefore, the subject of focus of the manuscript is an important one. The manuscript introduces (or rather tests) a new class of Lagrangian transport scheme CLaMS and compares its performance with the state of the art transport scheme popularly used by modern climate models. Using the same underlying dynamical fields, age tracers are advected by the two schemes and prominent differences are found in the performance of the two schemes. Care has been taken to isolate the influence of any other factors, as much as possible, in dictating the differences. Resultantly, the results show inter-representation differences in critical areas in the UTLS subtropics and higher latitudes using different transport metrics.

Therefore, these results from testing CLaMS in a practical setting with a realistic background stratospheric flow provides promise and strengthen confidence in the use of Lagrangian schemes in handling tracer advection. I do not doubt the scientific significance of this work and I believe researchers in the climate modeling community will definitely benefit from these results. I find the methodology scientific and the presentation fair. In this regard, I suggest the editor to **reconsider the manuscript pending major revisions.**

To facilitate the discussion process, I provide a couple of comments that require immediate attention and will definitely improve the quality of the manuscript. I follow up with a comprehensive review later.

**Comments :**

1. Since the authors themselves stress on the tracer being *passively* advected by the background flow, it would be great if the authors can include a figure showing the zonal mean Boreal summer and Austral summer climatology obtained from the model. This would greatly help the readers interpret the results that follow, in particular the meridional extent of the polar vortex.

2. Since the focus of the study is the UTLS region, I am surprised that none of the figures show the seasonally averaged tropopause profile. I urge the authors to add a thermal tropopause to Figures 1, 2, 4, 5, 6 and 7 in the manuscript.

3. In the same spirit as the comment above, since one of the schemes is formulated in purely isentropic cordinates in the stratosphere, where the vertical velocity represents the diabatic ascent rate - an important quantity for stratospheric mass upwelling - I strongly suggest the authors to demarcate the regions of tropical upwelling and extratropical downwelling by adding a line associated with $\dot{\theta} = 0$ in Figures 1, 2, 4, 5, 6 and 7.

4. The figure captions and the actual figures conflict with each other at a couple of places. In Figure 3 for instance, the relative and the absolute differences are flipped. Moreover, irrespective of the order, the (currrently (a)) relative differences suggest the relative difference shoud go negative at some point between 0.1 and 1 years, but that does not seem to be the case. The same holds for Figure 10 as well, where the Southern and northern hemisphere seem to flip. Due to this, I recommend the authors to revise Section 4.3 again.

5. To the extent of my understanding, Lagrangian schemes have been computationally more expensive than the traditional Eulerian schemes. It would be great if the authors can provide a rough estimate of the computation time used by the two schemes.

---

## Referee Comment (RC2) · Anonymous Referee #3 · 31 Aug 2020

Representation of air mass transport processes in the atmosphere, including in the upper troposphere and lower stratosphere (UTLS), is a long-standing question in the research community, as it is critically important to determine the global composition distribution and following climate projection. The choice of the transport scheme in climate models would induce the differences in efficiency of air mass transport and the tracer distribution. In this sense, a good representation of air mass transport in the numerical models is important for climate model simulations.

This study investigates the differences in the lower stratospheric transport between two transport schemes (EMAC-FFSL and EMAC-CLaMS) using the age spectrum, mean

age, and idealized tracers as diagnostics. The calculations are straightforward and the results appear reasonable, showing that a fully Lagrangian transport scheme results in significantly less numerical diffusion, stronger stratospheric transport barriers, and clearer structures in trace gas distributions. These results are noteworthy, and would definitely deepen our knowledge on the representation of air mass transport in UTLS and benefit the modeling community. The authors also discussed the underlying mechanisms and potential consequences from the choice of the transport scheme on simulations. In general, this work is interesting. The manuscript is well structured and written, and figures are nicely generated. I recommend publication after the following minor revisions.

Major points:

1) Abstract: I suggest the author to enhance the presentation here. They stated that they would assess the impact of the choice of trace gas transport scheme, however, what are the two schemes are not pointed out clearly (the Lagrangian scheme of EMAC-CLaMS and Eulerian scheme EMAC-FFSL?). Some statements here, for instance "In the lowermost stratosphere, air is much younger in EMAC (Line 11 – 12, Page 1)", introduced a little bit of confusion. What do you mean the "large-scale" and "smaller-scale" in Line 5-6? If my understanding is well, I would prefer to change "smaller-scale" to "regional-scale". What does the CCM mean? 2) Methods: In order to ensure the repeatability of the work, I would like to see the more details on the model and methods. For instance, what are the meteorological dynamical fields to force EMAC climate model? Could you provide the uppermost and lowest level of the 90 vertical layer of EMAC model configuration in this study? As to the Lagrangian modelling with EMAC-CLaMS, how the 300 million air parcels are initialized at the beginning of simulations? Are the air parcels are distributed uniformly in the atmosphere? 3) Section 3 Differences in the Zonal Mean State: Global Perspective: The authors diagnosed and compared many diagnostics here between two schemes, such as the mean age of air, chemical composition, inter-annual variability, and age spectrum shape. I am slightly confused as to these diagnostics. In my opinion, the paper might be more readable if the authors could add more explanations on the exact mean of these diagnostics. For example, what does the hotpot of inter-annual variability of mean age mean? Does its magnitude relate to the strength of air mass transport? What are the significance of the differences in this diagnostic between two transport schemes? Among those diagnostics, which one is of great importance to characterize the transport time scales? 4) Section 4 Differences in the representation of transport processes: One can notice that the variability in the age spectra with the transport time scale. There are multiple peak values or maxima occurred even within a period of 1-2 year. Could you provide more information on this issue? I would like to see the comparison of transport time spectra among three sub-regions as well (the tropical and mid-latitude stratosphere, the polar vortexes, and the lowermost stratosphere). For instance, why the shape of age spectra of three sub-regions are significantly different? One can see that the variabilities with transport time (year) is obvious in Figure 8 and 9, however, this feature is almost absent in Figure 10. Why? 5) Section 5 Discussion: The authors argued that reduced/enhanced numerical diffusive transportis responsible for the difference between the two transport schemes. The author might extend their discussion by providing explanations and emphasis the influence of this issue. For instance, as mentioned in the paper, the vertical velocity of EMAC-CLaMS is the diabatic heating rate, whereas EMAC-FFSL uses a kinematic vertical velocity, which could introduce, even in the UTLS region, larger difference in air mass transport (e.g., Hoppe, et al., 2016). Thus, I suggest the authors to discuss this point at end of Sect.5. 6) Conclusion: If possible, the paper could include simple comparisons to previous publications, for example, the study of Hoppe et al. (2014).

Minor points:

1) Line 15, P1: What does the acronyms CCM exactly mean? 2) Line 28, P2: A period is absent before "To our …". And Line 29, P2: A period could be deleted. 3) Line 13 P3: In section 2.1->In section 2. 4) Line 3 P7: Does the sentence"The Lagrangian

approach results in older air throughout most regions of the stratosphere"mean that more air mass could be transported to the stratosphere in the Lagrangian modelling? This is likely in agreement with previous studies. 5) Line 12, P5: It could be better to add some references here. 6) Line 30, P15: Figure 8 (a)->Figure 8 (b). 7) Line 21, P17: Figure 9 -> Figure 9 (b). 8) Line 4 P 18: Figure 9-> Figure 9 (a). 9) Line 5-6, P18: How did you conclude that "maxima in the spectra correspond to January-emitted tracers while minima correspond to July-emitted tracers"? Judging from the Figure 9 or other information?

---

## Author Response (AR1)

We thank the referees for their helpful, observant, and insightful comments. We believe the manuscript has been significantly improved through their contributions. Author responses address the comments above them. Some responses address multiple comments; in those cases the relevant comments numbers are listed.

**Anonymous Referee #1**:
**Major Comments**:

1. The issue of numerical diffusion in atmospheric transport is almost as old as climate modeling itself. In this regard, its discouraging that the Introduction does not discuss any historical perspective at all, given how central it is to the study. I strongly suggest the authors to add some discussion on it (atleast a paragraph; perhaps on Page 2 somewhere between Line 5 and 10). Almost every decade has some studies focused towards studying the effects of advection scheme on atmospheric transport. Some of these include :

   (a) Rood, R.B. (1987) : Numerical advection algorithms and their role in atmospheric transport and chemistry models. Reviews of Geophysics, 25(1), 71-100

   (b) Hall, T.M., Waugh, D.W., Boering, K.A. and Plumb, R.A. (1999) : Evaluation of transport in stratospheric models. Journal of Geophysical Research: Atmospheres, 104(D15), 18815-18839

   (c) Kent, J., Ullrich, P.A. and Jablonowski, C. (2014) : Dynamical core model intercomparison project: tracer transport test cases, Quarterly Journal of the Royal Meteorological Society, 140, 1279-1293.

   (d) Gupta, Aman, Edwin P. Gerber, and Peter H. Lauritzen (2020) : Numerical impacts on tracer transport: A proposed intercomparison test of Atmospheric General Circulation Models, Quarterly Journal of the Royal Meteorological Society

   This can help provide a big-picture perspective for the present manuscript as well, in concert with connecting it to the recent ECHAM/EMAC developments (which the authors have done quite well already).

   *Author Response*: We agree that including such a historical discussion would be a great addition to the introduction. We've expanded the following paragraph (after the one suggested) to include this.

2. Since the authors themselves stress on the tracer being passively advected by the background flow, it would be great if the authors can include a figure showing the zonal mean Boreal summer and Austral summer climatology obtained from the model. This would greatly help the readers interpret the results that follow, in particular the meridional extent of the polar vortex.

3. Since the focus of the study is the UTLS region, I am surprised that none of the figures show the seasonally averaged tropopause profile. I urge the authors to add a thermal tropopause to Figures 1, 2, 4, 5, 6 and 7 in the manuscript.

*Author Response* on items 2, 3: We have now included the zonal mean winds and monthly-average tropopause in all zonal mean figures. We agree with the referee; we believe that this information improves the intrepretation of results greatly.

4. In the same spirit as the comment above, since one of the schemes is formulated in purely isentropic cordinates in the stratosphere, where the vertical velocity represents the diabatic ascent rate - an important quantity for stratospheric mass upwelling - I strongly suggest the authors to demarcate the regions of tropical upwelling and extra-tropical downwelling by adding a line associated with $\dot{\theta} = 0$ in Figures 1, 2, 4, 5, 6 and 7.

   *Author Response*: We did try including lines showing $\dot{\theta} = 0$ in the zonal mean figures, but found that the additional information overcomplicated the figures and did not seem to improve interpretation; most of the $\dot{\theta} = 0$ contours were not smooth enough to be clearly meaningful, and those that were smooth were in rather expected locations (edges of the tropical pipe).

5. Section 4.3: I suggest the authors to please revisit this section. There are many typos and there is a mismatch in the order of plots and their description in this section (and in the figure caption). This has made it a bit tricky to navigate this section and interpret the results properly.

   *Author Response*: We have revised section 4.3 and corrected the error in the caption of Figure 10, referenced in the section.

**Minor comments**:

1. To the extent of my understanding, Lagrangian schemes have been computationally more expensive than the traditional Eulerian schemes. It would be great if the authors can provide a rough estimate of the computation time used by the two schemes.

   *Author Response*: We did wish to incorporate an estimate of the required resources (and thank the reviewer again for the early notification of the comment), but unforunately that calculation is no longer possible as the paralellized machine we used for calculations is being decomissioned, and the resources available on it are now drasitcally reduced. At the moment we do not have access to an available alternative. That being said, we have performed a comprehensive estimate of computation time used in a simulation, including a breakdown of time spent in each process.

2. P3 L29 : What was the range for hybrid-pressure coordinates. I only mention this for reproducibility sake.

*Author Response*: The vertical resolution of the model is identical to that used in Jöckel et al. 2016. We've added a note to that regard in the modeling section. For your curiosity, the higest EMAC model level is about 1 Pa.

3. P4 L5 : 3 million is a large number of particles. I have two points here. Firstly, can you mention (in the manuscript) how you arrived at this number or provide a reference for it? Moreover, has it been tested how changing this number can potentially affect the performance of the scheme? If this could potentially affect the performance of the advection scheme, particularly the eave structures, I consider acknowledging this point important.

4. P4 L6 : How sensitive is the parameterized mixing to the employed resolution? I believe it should not affect the results a lot. Can the authors comment on it a bit? (personally; not in the manuscript) Secondly, was this number fixed throughout the integration? If I understand correctly, depending on local mixing of parcels, the number of parcels can change over time, albeit slightly. If this is correct indeed, can you provide me an estimate of fractional change in this number throughout the interval of integration?

*Author Response* regarding 3 and 4: The number of air parcels in a CLaMS simulation is essentially the conuterpart to the resolution of Eurlerian grids; 3 million parcels is effectively fixed by the model setup, with the exception that the parcel count does vary slightly (plus or minus XXX%) over time. 3 million parcels is the "standard" resolution of the model. There is also a lower-resolution setting which is used for testing, but at this point only the high resolution setting is used for published results.Certainly changing the resoultion would have some changes on the transport characteristics of CLaMS. However, the conception of the CLaMS scheme requires that the Lyapunov exponent (the only other parameter controlling mixing besides resolution) be adjusted when the model resolution is changed. As both these parameters (resultion and the Lyapunov exponent) would be changed, it's difficult to say what changes on advection might occur, or even if those changes would be more significant than the changes in parcel locations throughout the simulation (a necessary consequence of alter model resolution). The choice of Lyapunov exponent and parcel count has been established since the introduction of three-dimensional mixing into CLaMS by Konopka et al. 2004 (cited in the same paragraph of the manuscrpt addressed by points 3 and 4).

5. P7 L6 : a deeper extent of the old polar vortex. Not sure what the authors mean by this.

*Author Response*: This refers to the fact that the age contours around the polar vortexes are lower in altitude in EMAC-CLaMS than EMAC-FFSL. We see now that the term "deeper" is confusing here as it could be interpreted as indicating lower altitudes or distance from the tropopause. We have edited the sentence for clarity.

6. P8 L16 : Can you please elaborate what you mean by recirculation differences? Also, I personally disagree that this is caused by differences in upper boundary conditions. If that were really the case, one would expect a seasonal cycle in this pattern and EMAC-CLaMS should have a younger air in the Northern Hemisphere in Figure 1(b) as well.

   *Author Response*: When material enters the tropical pipe it is lifted up into the deeper Stratosphere, turns over to higher latitudes, and sinks. As it sinks, it may mix with the tropical pipe, or might re-enter the tropical pipe via mixing in the lower Stratosphere followed by lifting into the pipe. "Recirculation" describes this process by which material can re-enter this circuit (over the pipe and the mid-latitudes). We are of the opinion that the results mentioned in this line are likely related to re-circulation, but we are not able to, with complete certainty, separate this effect from possible diffences in the upper boundary conditions. Owing to that, we mentioned the two possible causes and opted not to persue the topic further. However, we noticed that the sentence could be written more clearly, and have therefore modified it.

7. Figure 1 (b) and (e) are quite informative. I think the authors can also use the opportunity to connect the results of the numerics to the vanishing mixing at around 600K. More precisely, past studies (for instance Haynes and Shuckburgh 2000; https://doi.org/10.1029/2000. suggests that the tropical-subtropical mixing vanishes around 500-700K region (roughly). That the subtropical region at these isentropes show large difference between the two schemes is suggestive of the role numerical diffusion might play in (spurious) mixing between the two regions. This way the diffusion can be connected to resolved horizontal mixing in the stratosphere.

   *Author Response*: That's a great idea. We've added a sentence citing Haynes and Shuckberg 2000, and also cited the later work of Abalos, Legras, and Shuckberg 2016, which also calculated effective diffusivity and found similar results using reanalysis wind data.

8. P11 L1 : The results suggest.... How likely are differences in representation of transport barriers to be attributed to differences in parameterized mixing between the two schemes? Since the barrier (especially the subtropical barrier) manifest at relatively finer scales, can the mixing play a significant role in determining them?

   *Author Response*: Quantitative adjustments in mixing or resolution for gridded climate models (at least at contemporary resolutions) might not produce serious differences in cross-barrier mixing. However, our work decribes not a quantitative but qualitative difference in mxing between transport schemes. Whereas EMAC's resolution is fixed in space, the spatial resolution of CLaMS is strongest along transport barriers. This topic has been described in previous work on CLaMS, and is one of the strongest advantages of the model. Described shortly, CLaMS parcels tend to cluster around transport barriers, and due to this clustering mixing occurs frequently at the edges of transport barriers. It may be the case that even finer CLaMS resolutions (i.e. additoinal parcels)

would resolve even sharper features around transport barriers, perhaps providing an improved representation of mixing at these locations as well. This aside, the qualitative difference in mixing and resolution between EMAC-FFSL and EMAC-CLaMS is, in our view, a probable cause of the differences in cross-barrier transport diagnostics between the two schemes. Furthermore, the design of our study is such that no other plausible explanation is available for horizontal barriers, because the wind fields used in both models are identical (although, as noted in the text, the fields are not applied in identical ways in the transport schemes).

9. P14 L13 : It would help if you can provide a one sentence description as to why the ratio of moments is, as you state, a critical parameter.

   *Author Response*: The studies we cited in that paragraph use a prescribed value for the ratio of moments. This is necessary for constraining the age spectrum, so that inverse methods can be applied to estimate it. We have extended that sentence, adding more detail on the topic.

10. P17 L5-6 : It would be great if the authors can provide some insight into why this is.

    *Author Response*: Although we do agree that a more detailed explanation may be helpful, we are, unfortunately, uncertain of how to concisely provide that material while avoiding the loss of focus on the topic at hand in that paragraph.

**Figures**: A general comment on the figures: The figure captions and the actual figures conflict with each other at a couple of places. The same holds for Figure 10 as well, where the Southern and northern hemisphere seem to flip. Due to this, I recommend the authors to revise Section 4.3 again. The figures quality is quite good overall. However, I would suggest the authors to increase the xlabel, ylabel, xtick, ytick size for all the figures - especially the contour-fill plots, to make the figures look a bit more appealing.
*Author Response*: There was indeed an error in the caption of Figure 10, which has now been corrected. The text of section 4.3 has also been revised.

1. Figure 2 : b and e instead of b and c. Also, why does the northern polar region (400-550K) in 2 (b) (color) look qualitatively different when compared to Figure 1 (b). This does not happen in any other region.

   *Author Response*: The correction has been applied. In terms of the qualitative difference in the northern hemisphere between 2b and 1b, we are also somewhat surprised by the difference. That something similar to this might occur is not surprising for us; the forward tracers "focus" on the lower stratosphere by reducing the role of older air, compared to mean age of air. The effect of that can be seen in the slightly lower altitudes of the locations of greatest inter-scheme differences (comparing Figure 1b/e

and 2b/e), but this effect is seemingly much more pronounced in the northern winter hemisphere. The cause may be indicated by the considerable fraction of young (¡2 years, the lifetime of this particular forward tracer) air in the northern polar vortex (Figure 9a). This fraction is much greater than the case of the southern polar vortex (Figure 9b). This may explain why the northern winter polar region is so much more sensitive to the choice of diagnositic (mean age or forward tracer) than other regions seem to be. This difference can also be seen in the northern and southern midlatitudes (Figure 8a, 8c), where again the north shows a stronger fraction of young (¡2 years) air than the south, although to a weaker extent than seen in the polar vortexes.

2. Figure 3 : The relative and the absolute differences are flipped, as per the caption. Moreover, irrespective of the order, the (currrently (a)) differences suggest that the relative difference shoud not go negative at some point between 0.1 and 1 years, but that does not seem to be the case i.e. The blue solid and dashed curves intersect in (a), while they do not in (b) - so something seems to be wrong here.

   *Author Response*: The caption has been adjusted to incorporate the correction and for clarity. The relative difference referred to by the caption is the absolute difference normalized by EMAC-CLAMS, so differences of 0% indicate the two are identical.

3. Figure 4 : The contour intervals are not mentioned (which is fine), but are the contour labels for the horizontal gradients equally spaced? Again, I would request adding the thermal tropopause here.

   *Author Response*: We wrote the figure caption with the aim of clarifying this point in particular, but have revised the caption with the hopes of greater clarity. The contours, both the shading and the lines, are equally spaced.

4. Figure 4 : Why are the southern hemisphere winter barriers stronger? This has not been discussed in the main text.

   *Author Response*: We assume that the question addresses the southern polar vortex. The southern polar vortex, in comparison to the northern polar vortex, is colder and therefore has a stronger meridional temperature gradient. This makes the southern polar vortex winds stronger than those of the northern counterpart, and a stronger wind gradient between the southern polar vortex and southern midlatitudes compared to the gradient in the northern hemisphere. This in turns means that the PV gradient in the south in stronger than in the north. Strong PV gradients are associated with strong transport barriers, so this results in a stronger transport barrier for the southern polar vortex. We have added a short mention of the topic at the begging of section 4.2.

5. Figure 5 : Standard deviation

   *Author Response*: The correction has been applied.

6. Figure 7 : You mean to those of Figure 2?

   *Author Response*: Yes we do, the correction has been applied.

7. Figure 8 : Missing xlabels. Moreover, are you suggesting that all of the 0.05% of the red curve is due to diffusion? Also there is a mismatch in the figure caption and the figure reference on P15 L27 as Figure 8(a) shows the spectrum for the summer midlatitudes.

   *Author Response*: The figure reference has been corrected. We have checked Figure 8, but it seems that the xlabels are present. With regards to the 0.05% of the red curve, we are not entirely sure that we understand the question, If we are correct in the assumption that the question refers to the 3-month-old component of the age spectra (mentioned on P17 L3), then perhaps this clarification is helpful: EMAC-CLaMS shows no air of this extremely young age in any of the age spectra shown in Figure 8, whereas EMAC-FFSL shows some small amount of air of this age. This difference must be due to a difference between the two schemes. By the construction of the experiment, advection processes should be similar between the two schemes, whereas diffusion is certainly different. The presence of this extremely young air in EMAC-FFSL is therefore due to stronger diffusion in that scheme. However, the presence of this air is certainly not only due to diffusion. Rather, our interpretation is that diffusion carries air over regions of slow transport (the tropopause) while advection (and diffusion) carries this air throughout the stratosphere. In other words, we are not suggesting that diffusion is the only relevant process in the transport schemes, but rather that excessive diffusion in EMAC-FFSL creates a bias in results towards faster transport (in comparison to EMAC-CLaMS).

**Typographical errors and grammatical suggestions**: these comments have simply been adopted.
**Anonymous Referee #3**
**Major points**:

1. Abstract: I suggest the author to enhance the presentation here. They stated that they would assess the impact of the choice of trace gas transport scheme, however, what are the two schemes are not pointed out clearly (the Lagrangian scheme of EMAC-CLaMS and Eulerian scheme EMAC-FFSL?). Some statements here, for instance In the lowermost stratosphere, air is much younger in EMAC (Line 11 12, Page 1), introduced a little bit of confusion. What do you mean the large-scale and smaller-scale in Line 5-6? If my understanding is well, I would prefer to change smaller-scale to regional-scale. What does the CCM mean?

*Author Response*: We agree that the transport schemes should be identified earlier, and have added short descriptors for the two schemes in the second sentence of the abstract. We prefer to use the term "smaller-scale" over "regional-scale" because we use the term "region" to describe parts of the stratosphere which describe large areas thereof. We have also changed the wording where the "younger" was used to specify mean age of air. CCM means "Chemistry-Climate Model". We used that term erroneously, and have corrected it.

2. Methods: In order to ensure the repeatability of the work, I would like to see the more details on the model and methods. For instance, what are the meteorological dynamical fields to force EMAC climate model? Could you provide the uppermost and lowest level of the 90 vertical layer of EMAC model configuration in this study? As to the Lagrangian modelling with EMAC-CLaMS, how the 300 million air parcels are initialized at the beginning of simulations? Are the air parcels are distributed uniformly in the atmosphere?

*Author Response*: The model is free-running; no meteorological dynamical fields are used for forcing (unless sea surface temperatures are counted among those). The CLaMS parcels are initialized in a uniform distribution, but the uniformity decomposes rapidly, and after the ten-year spin-up period there is certainly little if any trace remaining of the initial distribution. For greater clarification on where readers can seek more information about the model set-up, we have added a sentence citing Jöeckel et al 2016. Additonally, we have added a note clarifying that the EMAC was free-running.

3. Section 3 Differences in the Zonal Mean State: Global Perspective: The authors diagnosed and compared many diagnostics here between two schemes, such as the mean age of air, chemical composition, inter-annual variability, and age spectrum shape. I am slightly confused as to these diagnostics. In my opinion, the paper might be more readable if the authors could add more explanations on the exact mean of these diagnostics. For example, what does the hotpot of inter-annual variability of mean age mean? Does its magnitude relate to the strength of air mass transport? What are the significance of the differences in this diagnostic between two transport schemes? Among those diagnostics, which one is of great importance to characterize the transport time scales?

*Author Response*: High variability in mean age is associated with high varibility in dynamics. Seeing stronger variability in one transport scheme's results indicates that the scheme produces greater "effective" variability in dynamics (dynamics are identical between the two schemes), in principle due to a different characterization of diffusion between the models. In our view, the diagnostic with the greatest relevance for characterizing transport time scales is the mean age of air, chiefly due to its relative simplicity. Other diagnostic provide useful insights of either a supplementary charcter (2-year forward tracers, for example) or information which is more relevant to research involving

more specific topics (Figure 7's ratio of moments, for example, are surely of substantial interest to those studying the use of age spectra).

4. Section 4 Differences in the representation of transport processes: One can notice that the variability in the age spectra with the transport time scale. There are multiple peak values or maxima occurred even within a period of 1-2 year. Could you provide more information on this issue? I would like to see the comparison of transport time spectra among three sub-regions as well (the tropical and mid-latitude stratosphere, the polar vortexes, and the lowermost stratosphere). For instance, why the shape of age spectra of three sub-regions are significantly different? One can see that the variabilities with transport time (year) is obvious in Figure 8 and 9, however, this feature is almost absent in Figure 10. Why?

   *Author Response*: We are not certain what age spectra are being referenced in the first three sentences of this comment. Local (annual) maxima occur in age spectra for the times corresponding to winter transport, when tropical BDC upwelling is strongest. Regarding multiple maxima over periods of a year, the only example we clearly see is Figure 8a, and only for EMAC-FFSL. In that case, the high variability at one year of age reduces the relevance of the slight local minima at that point; this particular feature of this particular age spectra may be merely a coincidence. Regarding the shapes of age spectra from various regions, we believe that this topic has already been covered by other studies focusing on age spectra, and would prefer to keep the focus of our study on the diffrences between the two transport schemes.

5. Section 5 Discussion: The authors argued that reduced/enhanced numerical diffusive transportis responsible for the difference between the two transport schemes. The author might extend their discussion by providing explanations and emphasis the influence of this issue. For instance, as mentioned in the paper, the vertical velocity of EMAC-CLaMS is the diabatic heating rate, whereas EMAC-FFSL uses a kinematic vertical velocity, which could introduce, even in the UTLS region, larger difference in air mass transport (e.g., Hoppe, et al., 2016). Thus, I suggest the authors to discuss this point at end of Sect.5.

   *Author Response*: P4 L15-17 mentions this point and notes the possible difference. While this difference is present between the two schemes, the two vertical velocities are both calculated from EMAC and are therefore consistent. Because of this, the differences in vertical advection between the two schemes caused by the kind of vertical velocity used should be small compared to the differences caused by the different applications of these winds (due to the interpolation of wind velocities onto CLaMS parcels) and by the differences in vertical diffusion which are a more fundamental difference in Lagrangian and Eulerian schemes.

6. Conclusion: If possible, the paper could include simple comparisons to previous publications, for example, the study of Hoppe et al. (2014).

   *Author Response*: We agree. We have now included a sentence mentioning the work of Hoppe et al. and Stenke et al. in the conclusions.

**Minor points**:

1. Line 15, P1: What does the acronyms CCM exactly mean?

   *Author Response*: Chemistry climate model. The term was used in error in that line, and the term has been changed to "chemstry-climate".

2. Line 28, P2: A period is absent before To our . . .. And Line 29, P2: A period could be deleted.

   *Author Response*: The correction has been applied.

3. Line 13 P3: In section $2.1 -> $ In section 2.

   *Author Response*: The correction has been applied.

4. Line 3 P7: Does the sentence "The Lagrangian approach results in older air throughout most regions of the stratosphere" mean that more air mass could be transported to the stratosphere in the Lagrangian modelling? This is likely in agreement with previous studies.

   *Author Response*: For a given point in the atmosphere $\mathbf{x}$, the mean age of air is the average time required to transport air from the tropical surface to $\mathbf{x}$. Increases of mean age of air indicate that this transport time becomes longer i.e. that transport becomes slower. So the meaning of this sentence is that transport is slower in the Lagrangian perspective. We are not aware of previous studies that find Lagrangian transport to be faster than Eurlerian. On the contrary, the works of Hoppe et al. 2014 and 2016 and of Stenke et al. 2008 and 2009 - to our knowledge the only other works that have performed such a comparison - all found that transport was weaker in the Lagrangian perspective than in the Eulerian perspective.

5. Line 12, P5: It could be better to add some references here.

   *Author Response*: We've added a citation to a study where longer age spectra were used.

6. Line 30, P15: Figure 8 (a) − > Figure 8 (b).

7. Line 21, P17: Figure 9 − > Figure 9 (b).

8. Line 4 P 18: Figure 9 − > Figure 9 (a).

*Author Response* regarding 6, 7, 8: The corrections have been applied.

9. Line 5-6, P18: How did you conclude that maxima in the spectra correspond to January-emitted tracers while minima correspond to July-emitted tracers? Judging from the Figure 9 or other information?

*Author Response*: The assignment of months to the maxima and minima is a trivial (we mean this in the mathematical sense - it's not necessarily obvious) conclusion from the nature of the x-axes of the age spectra in Figure 9. Figure 9a is dated in January and the maxima occur on ages 1, 2, 3, etc. years, therefore during January months of 1, 2, 3, etc. years before. In contrast Figure 9b is dated in July and shows maxima occuring at ages 2.5, 3.5, etc., therefore also during the January months of 2, 3, etc. years ago.

[revised manuscript text omitted]

---

## Author Response (AR2)

Two changes were made to the document.

1. A note on code availability was added.

2. The word "and" was added to the list of names in the first sentence of the acknoledgement section for clarity.